# Online Black-Box Prompt Optimization with Regret Guarantees under Noisy Feedback

**Jinjie Fang**[1]  **Runwen You**[1]  **Wanli Shi**[1]  **Wenkang Wang**[1]
**Ganyu Wang**[2]  **Haozhen Zhang**[1]  **Yi Chang**[1,3,4†]  **Bin Gu**[1†]

[1]School of Artificial Intelligence, Jilin University, China
[2]Western University, Canada
[3]International Center of Future Science, Jilin University, China
[4]Engineering Research Center of Knowledge-Driven Human-Machine Intelligence, MOE, China
`{wanli_sh, yichang, gubin}@jlu.edu.cn {gwang382}@uwo.ca`
`{fangjj24, runwen25, wangwk25, haozhen23}@mails.jlu.edu.cn`

## Abstract

Generative AI excels in various tasks through advanced language modeling techniques, with its performance heavily influenced by input prompts. This has driven significant research into prompt optimization, particularly in commercial generative AI platforms, where prompt optimization is treated as a black-box optimization problem. Most existing research on black-box prompt optimization primarily focuses on offline learning and overlooks the randomness in outputs. However, in real-world applications, black-box prompt optimization typically operates in an online learning setting, which remains largely unexplored, especially given the noisy outputs. To address these challenges, we propose an **A**daptive **O**nline **Z**eroth-order **P**rompt **T**uning (AOZPT) approach which integrates zeroth-order optimization with online learning in the non-convex setting. Specifically, we developed an uncertainty-scale-adjustment mechanism to mitigate the noise inherent in generative AI and the high variance associated with zeroth-order estimates. We conducted a comprehensive regret analysis of the AOZPT approach, and the results indicate that sublinear regret convergence is achievable. Extensive generative experiments demonstrate that AOZPT outperforms existing black-box prompt tuning methods, particularly in terms of stability in online scenarios.

## 1 Introduction

Generative artificial intelligence (AI) leverages advanced contextual understanding and language modeling techniques to excel across a wide range of tasks (Feuerriegel et al., 2024; Brynjolfsson et al., 2023; Epstein et al., 2023). These capabilities facilitate the generation of high-quality text, code, and multimodal content, with applications in financial analysis, medical diagnosis support, and automated content creation (Li et al., 2023; Zhou et al., 2024; Ji et al., 2024). The generative process is partially influenced by the model's inherent randomness, which arises from random sampling, non-deterministic training elements, and variations in random seed initialization (Das & Varshney, 2022; Liu et al., 2024a; Gandee et al., 2024). These mechanisms enhance flexibility, allowing the model to generate diverse and creative content across various tasks and contexts.

Generative AI achieves diverse functionalities primarily through fine-tuning (FT) or prompt tuning (PT). FT involves adjusting all model weights to optimize performance for specific tasks; however, it demands substantial computational resources, large datasets, and often leads to reduced generalization and increased deployment complexity (Kenton & Toutanova, 2019; Liu, 2019; Liu et al., 2021). In contrast, PT updates only a small subset of parameters, significantly reducing computational and data requirements while preserving the model's inherent knowledge and adaptability (Lester et al., 2021; Zhang et al., 2024; Gao et al., 2020). Traditional white-box prompt tuning methods rely on access to a model's intermediate representations (Liu et al., 2021; Li & Liang, 2021; Zhou et al., 2022), while

---

[†]Corresponding Author.

black-box prompt tuning becomes essential when intermediate representations are inaccessible (Sun et al., 2022; Diao et al., 2022; Cheng et al., 2023; Liu et al., 2024b; Wu et al., 2024). Notably, black-box prompt tuning enables the optimization of input prompts without requiring a detailed understanding of the model's internal mechanisms.

Current research on black-box prompt tuning predominantly focuses on offline scenarios using pre-established datasets. For example, Sun et al. (2022) proposed BBT, an offline method that optimizes continuous prompts in a low-dimensional subspace using random projection and derivative-free optimization techniques. Similarly, Deng et al. (2022) introduced RLPROMPT, which employs reinforcement learning to optimize discrete text prompts within an offline framework. Furthermore, Diao et al. (2022) presented BDPL, an offline method for adapting large pre-trained language models through the optimization of discrete prompts without accessing model parameters. For gradient-based optimization, Zhan et al. (2024) developed the Zeroth-Order Tuning algorithm, designed for offline black-box prompt tuning using inference APIs exclusively. Additionally, Zhang et al. (2024) proposed a zeroth-order prompt tuning framework that addresses high-dimensional prompt optimization challenges in offline settings through subspace learning and selection strategies. Hu et al. (2024) introduced the ZOPO method, designed for offline learning scenarios, which effectively optimizes discrete prompts through input domain transformation, NTK-GP-enhanced derivative-free optimization, and uncertainty-informed local exploration. Collectively, these methods demonstrate flexibility and strong performance, providing effective solutions for offline black-box prompt tuning.

Offline black-box prompt tuning methods lack adaptability to dynamic data changes, posing a significant limitation for applications that require real-time updates. For example, in real-time customer support systems, online learning dynamically refines prompts based on ongoing user interactions, improving response accuracy and relevance (Upadhyaya, 2024). Similarly, in e-commerce platforms, online learning analyzes user browsing behavior in real time to adjust recommendation content, providing more personalized and precise services (Nkwo et al., 2018). In such scenarios, which demand real-time interaction or feedback, offline black-box prompt optimization is often ineffective or impractical. In contrast, online learning continuously optimizes prompts by integrating streaming data, enabling systems to dynamically adapt to evolving information. As a result, online black-box prompt tuning is more suitable for real-time applications, particularly those requiring rapid responses and dynamic adjustments, demonstrating substantial potential for practical implementation.

Nevertheless, implementing black-box prompt tuning for generative AI in online learning contexts presents notable challenges. First, the inherent randomness in generative AI models, while beneficial for enhancing content diversity, is often perceived as noise. This noise introduces output uncertainty, complicating prompt optimization in online black-box scenarios. Second, conventional black-box prompt optimization techniques, such as Bayesian optimization (Shahriari et al., 2015) or evolutionary algorithms (Bartz-Beielstein et al., 2014), require frequent surrogate model updates or the evaluation of a large number of samples. These requirements render them impractical for online learning scenarios (Sun et al., 2022; Zhang et al., 2024; Chen et al., 2023; Zhao et al., 2023; Guo et al., 2023; Lange et al., 2024). In contrast, gradient estimation-based methods, particularly zeroth-order optimization (ZOO), offer a more efficient, flexible, and robust framework for online black-box prompt tuning (Zhan et al., 2024; Hu et al., 2024; Zhang et al., 2024). However, ZOO approximates gradients using a limited number of function evaluations, often leading to high variance during the search process (Gu et al., 2016; Liu et al., 2018; Feng & Wang, 2023). This variance further exacerbates uncertainty in optimization, increasing its complexity.

To address the challenges of noise from generative AI and high variance in zeroth-order estimates in online black-box prompt optimization, we propose **A**daptive **O**nline **Z**eroth-order **P**rompt **T**uning (AOZPT), the first method to combine black-box prompt tuning with online learning. In simulated streaming data scenarios, AOZPT continuously adjusts prompts for generative AI based on incoming data, maintaining optimal performance throughout the learning process. Furthermore, to mitigate uncertainties arising from zeroth-order variance and genera-

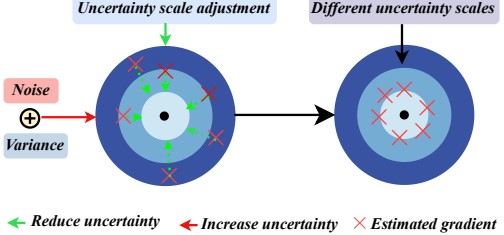

Figure 1: The adaptive uncertainty scale adjustment mechanism.

tive AI noise, we incorporate an adaptive uncertainty scaling mechanism (Figure 1) into the update process, effectively reducing gradient uncertainty.

The key contributions are summarized as follows:

- This paper proposes the AOZPT approach, the first to integrate black-box prompt tuning with online learning. AOZPT dynamically optimizes prompts based on streaming data, maintaining optimal performance throughout continuous learning.

- The AOZPT approach incorporates an adaptive uncertainty scaling mechanism to mitigate the noise in outputs of generative AI and the high variance arising from zeroth-order gradient estimates.

- We present a formal regret analysis of AOZPT in non-convex settings, demonstrating that sublinear regret convergence is achievable. Additionally, we evaluate the AOZPT method on both text-to-text and text-to-image tasks, with results consistently showing that AOZPT outperforms baseline models.

## 2 RELATED WORKS

### 2.1 WHITE-BOX AND BLACK-BOX PROMPT TUNING

Prompt tuning, a powerful paradigm originating in natural language processing, has recently gained significant attention. This approach focuses on designing and optimizing prompts to adapt models for diverse downstream tasks. Early efforts in prompt tuning relied on manually crafted prompts to guide language models toward desired outputs (Petroni et al., 2019). However, this method is both time-intensive and resource-demanding (Jiang et al., 2020). To address these challenges, researchers developed automatic prompt tuning techniques, which optimize prompts by learning effective representations (Shin et al., 2020). Automatic prompt tuning can be broadly categorized into two types: white-box and black-box prompt tuning. White-box prompt tuning assumes full access to the model, enabling direct interaction with its parameters and gradients (Li & Liang, 2021; Liu et al., 2021; Lester et al., 2021). Conversely, when access to a model's internal mechanisms is restricted—such as when a language model is provided as a service through an API—black-box prompt tuning becomes necessary. Recent advancements in black-box prompt tuning have introduced methods such as reinforcement learning (Deng et al., 2022), policy gradient (Diao et al., 2022), and genetic algorithms (Zhang et al., 2024; Sun et al., 2022). These methods are highly versatile, accommodating a wide range of tasks and models without requiring any modifications to the underlying model architecture.

### 2.2 ONLINE NONCONVEX LEARNING

Online learning is a paradigm where models are continuously updated in response to new data, as opposed to being trained in batch mode on static datasets. Traditional approaches to online learning have primarily relied on shallow models to address convex optimization problems. However, recent research has increasingly focused on non-convex scenarios. For instance, Hazan et al. (2017) introduced the concept of local regret as an alternative to traditional regret analysis in non-convex online learning. Unlike the standard regret used in online convex optimization, local regret is confined to a sliding window, making it "local" in nature. Aydore et al. (2019) extended this concept by proposing dynamic local regret to address concept drift in data streams. Their method incorporates an exponential average over the sliding window of local regret and leverages past gradients within the window, enhancing computational efficiency. Gao et al. (2020) presented an online normalized gradient descent algorithm for cases where gradient information is available and a bandit online normalized gradient descent algorithm for scenarios where only loss function values can be accessed. Additionally, Roy et al. (2019) explored the application of Gaussian Bandit Gradient Descent to online non-convex optimization. Kaya et al. (2023) proposed a communication-efficient zeroth-order distributed online optimization algorithm, which integrates an error feedback mechanism with a federated learning framework to enable multi-agent target tracking and optimization in communication-constrained environments. Most recently, Hua et al. (2024) proposed a residual feedback-based single-point distributed online non-convex optimization algorithm.

## 3 METHOD

### 3.1 ONLINE BLACK-BOX PROMPT OPTIMIZATION

**Online black-box prompt tuning:** In an online learning scenario, a stream sample $\xi^t$ is received at each round $t = 0, ..., T - 1$, comprising an input sentence $x^t$ and its corresponding true label $y^t$, i.e., $\xi^t = (x^t, y^t)$. Let $\mathcal{G}$ represent the black-box generative model and $\ell$ denote the loss function. The online black-box prompt tuning task involves minimizing the objective function $f^t$ by optimizing the prompt $\phi$:

$$f^t \left( \phi^t \right) \triangleq \ell \left( \mathcal{G} \left( \phi^t; x^t \right), y^t \right). \tag{1}$$

Based on the preceding discussion, mainstream black-box optimization methods, such as Bayesian and evolutionary algorithms, are impractical in online learning scenarios, necessitating gradient-based methods. However, directly applying gradient-based methods to optimize $\phi$ presents challenges, as $\phi$ represents a natural language sentence involving numerous discrete structures, rendering gradient-based methods unsuitable.

**In-context Learning Prompt Generator** To address the challenge of optimizing discrete prompts with gradient-based methods, we employ the INSTRUCTZERO framework for prompt generation (Chen et al., 2023). Within this framework, we optimize a low-dimensional continuous vector $z^t \in \mathbb{R}^d$, referred to as a soft prompt, to generate a high-quality discrete semantic instruction $\phi^t$, known as a hard prompt. Specifically, we represent a frozen open-source LLM as $\mathcal{F}$ and use a random projection matrix $A \in \mathbb{R}^{D \times d}$ ($D \gg d$) to project the low-dimensional vector $z^t \in \mathbb{R}^d$ into the high-dimensional embedding space $\mathbb{R}^D$ of $\mathcal{F}$. The resulting concatenated embedding is then input into $\mathcal{F}$ for generating semantic prompts. This process can be mathematically expressed as follows:

$$\phi^t = \mathcal{F} \left( A z^t + \phi_0; \xi^t \right). \tag{2}$$

This approach simplifies the process and enhances flexibility by optimizing soft prompts, represented as low-dimensional continuous vectors, instead of directly optimizing discrete hard prompts. Additionally, it effectively leverages the LLM's contextual understanding capabilities, facilitating the generation of high-quality prompts.

### 3.2 ADAPTIVE UNCERTAINTY SCALE ADJUSTMENT MECHANISM

The implementation of black-box prompt tuning for generative AI in online learning scenarios poses significant challenges. First, the intrinsic output noise of generative AI models generates unstable outputs, introducing uncertainty into the optimization process. Second, zero-order methods rely on limited function evaluations to approximate gradients, often resulting in high variance.

**Noise of generative AI output:** The output of the generative AI is often accompanied by randomness, even with fixed model parameters and inputs, the outputs may still vary. We define the randomness as $\delta \left( z^t \right)$, and the objective function with randomness can be defined as:

$$f_\delta^t \left( z^t \right) \triangleq f^t \left( z^t \right) + \delta \left( z^t \right). \tag{3}$$

**High variance of zeroth-order optimization:** ZOO estimates function gradients by sampling random perturbations within the domain and analyzing the resulting changes in output, providing a flexible framework for gradient estimation in black-box scenarios (Shamir, 2017). However, zero-order methods, which rely on a limited number of function evaluations for gradient approximation, often suffer from high variance during the search process (Liu et al., 2018). In the context of prompt tuning for generative AI, the inherent noise in the model's output renders this gradient estimation process a noisy zeroth-order approximation. To compute the partial derivative with respect to the soft prompt $z$, we utilize the noisy central two-point random gradient estimator:

$$\hat{\nabla}_z f_\delta^t \left( z^t \right) = \frac{f_\delta^t \left( z^t + \mu u^t \right) - f_\delta^t \left( z^t - \mu u^t \right)}{2\mu} u^t, \tag{4}$$

where $\mu$ is the smoothing parameter, and $u$ is the direction vector sampled from the unit sphere $\mathcal{S}^d := \left\{ u \in \mathbb{R}^d : \|u\|_2 = 1 \right\}$.

**Adaptive uncertainty scale adjustment:** To address the uncertainty caused by the noise in generative AI and the variance in zeroth-order estimates, we introduce an adaptive uncertainty scaling mechanism.

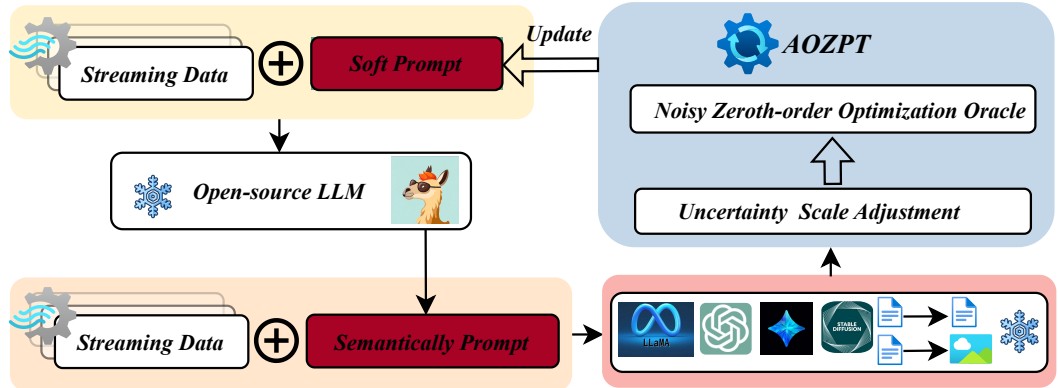

Figure 2: The architecture diagram of AOZPT model.

This mechanism incorporates the exponentially weighted moving average of squared gradients into the update process, effectively reducing gradient uncertainty. We define the gradient update as follows:

$$z^{t+1} \leftarrow z^t - \eta \cdot \frac{\mathbf{m}_t}{\sqrt{\mathbf{v}_t + \epsilon}}. \tag{5}$$

Here, $\mathbf{m}_t$ can be interpreted as a "momentum", incorporating the exponentially weighted moving average of historical gradients to facilitate smoother and more stable gradient updates. The term $\mathbf{m}_t$ is defined as follows:

$$\mathbf{m}_t = \frac{1}{W} \sum_{i=0}^{w-1} \alpha^i \cdot \hat{\nabla}_z f_\delta^{t-i} \left( z^{t-i} \right), \tag{6}$$

and $\mathbf{v}_t$ can be regarded as an "adaptive term", incorporating the exponentially weighted moving average of squared historical gradients. Including this term in the denominator enables the scaling of estimated gradients across dimensions, effectively balancing gradient magnitudes and reducing overall uncertainty. The term $\mathbf{v}_t$ is defined as follows:

$$\mathbf{v}_t = \frac{1}{M} \sum_{i=0}^{w-1} \beta^i \cdot \left[ \hat{\nabla}_z f_\delta^{t-i} \left( z^{t-i} \right) \right]^2, \tag{7}$$

where $0 < \alpha, \beta < 1$, and the superscript $i$ of the $\alpha^i$ and $\beta^i$ indicates the exponent to assign more weights to the most recent values; $W = \sum_{i=0}^{w-1} \alpha^i$ and $M = \sum_{i=0}^{w-1} \beta^i$ serve as the normalization parameter for the exponential average, ensuring that $\frac{1}{W} \sum_{i=0}^{w-1} \alpha^i = 1$ and $\frac{1}{M} \sum_{i=0}^{w-1} \beta^i = 1$; $f_\delta^t \left( z^t \right) = 0$ for $t \leq 0$.

### 3.3 ADAPTIVE ONLINE ZEROTH-ORDER PROMPT TUNING

The AOZPT approach optimizes prompts in online black-box scenario (Figure 2). During the prompt generation phase, we utilize a frozen open-source LLM for instance optimization to refine the prompt tuning. This approach capitalizes on the LLM's robust capabilities in contextual learning and language comprehension. Specifically, we leverage the model's deep understanding of linguistic patterns and context to generate high-quality, semantically rich prompts by optimizing its soft prompts. In the prompt update phase, we introduce perturbations to the soft prompts to compute the differential of the output loss function, thereby approximating the gradient using zeroth-order gradient estimation. Additionally, we incorporate an adaptive uncertainty scale adjustment mechanism to address the uncertainty of online black-box prompt tuning (Algorithm 1).

---

**Algorithm 1** AOZPT

---

**Input:** learning rate $\eta$, smooth parameter $\mu$, the length of the sliding window $w$, weighting parameter $\alpha$ and $\beta$, normalization parameter $W$ and $M$, a small constant $\epsilon$, initialize $w$-dimensional zero-initialized gradient vector $\Lambda$.

**Output:** $\{z^t\}_{t=1}^T$.

Initialize soft prompt $z^0$.

**for** $t = 0$ **to** $T - 1$ **do**

    Receive $\xi^t = \{x^t, y^t\}$.

    Get $u^t$ by sampled from unit sphere $\mathcal{S}^d$.

    Compute: $\phi_+^t = \mathcal{F}\left(\mathrm{A}\left(z^t + \mu u^t\right) + \phi_0; \xi^t\right)$ and $\phi_-^t = \mathcal{F}\left(\mathrm{A}\left(z^t - \mu u^t\right) + \phi_0; \xi^t\right)$.

    Compute $f_\delta^t\left(z^t + \mu u^t\right)$ and $f_\delta^t\left(z^t - \mu u^t\right)$:

$$f_\delta^t\left(z^t + \mu u^t\right) = \ell\left(\mathcal{G}\left(\phi_+^t; x^t\right), y^t\right) + \delta\left(z^t + \mu u^t\right),$$
$$f_\delta^t\left(z^t - \mu u^t\right) = \ell\left(\mathcal{G}\left(\phi_-^t; x^t\right), y^t\right) + \delta\left(z^t - \mu u^t\right).$$

    Compute the estimation gradient $\hat{\nabla}_z f_\delta^t\left(z^t\right)$:

$$\hat{\nabla}_z f_\delta^t\left(z^t\right) = \frac{f_\delta^t\left(z^t + \mu u^t\right) - f_\delta^t\left(z^t - \mu u^t\right)}{2\mu} u^t$$

    Update gradient vector:

$$\Lambda = \left[\hat{\nabla}_z f_\delta^{t-w+1}\left(z^{t-w+1}\right), \hat{\nabla}_z f_\delta^{t-w+2}\left(z^{t-w+2}\right), ..., \hat{\nabla}_z f_\delta^t\left(z^t\right)\right]$$

    Compute $\mathbf{m}_t \leftarrow \frac{1}{W}\sum_{i=0}^{w-1}\alpha^i \cdot \hat{\nabla}_z f_\delta^{t-i}\left(z^{t-i}\right)$ and $\mathbf{v}_t \leftarrow \frac{1}{M}\sum_{i=0}^{w-1}\beta^i \cdot \left[\hat{\nabla}_z f_\delta^{t-i}\left(z^{t-i}\right)\right]^2$.

    Update $z^{t+1} \leftarrow z^t - \eta \cdot \frac{\mathbf{m}_t}{\sqrt{\mathbf{v}_t} + \epsilon}$.

**end for**

---

## 4 ANALYSIS

### 4.1 DEFINITIONS

**Definition 4.1. Local regret for online non-convex optimization:** The sliding window mechanism provides an effective means of evaluating online learning algorithms by calculating the exponentially weighted moving average of the loss, assigning greater weight to more recent losses (Hazan et al., 2017). The exponentially weighted sliding-window average function defined as follows:

$$F_{w,\alpha}^t\left(z^t\right) \triangleq \frac{1}{W}\sum_{i=0}^{w-1}\alpha^i \cdot f^{t-i}\left(z^{t-i}\right). \tag{8}$$

The local regret for online black-box prompt tuning is formally defined by the accumulated squared norm of the gradient of the exponentially weighted sliding-window average (Aydore et al., 2019):

$$\mathfrak{R}(T) \triangleq \sum_{t=1}^T\left\|\nabla_z F_{w,\alpha}^t\left(z^t\right)\right\|_2^2, \tag{9}$$

where $\nabla_z F_{w,\alpha}^t\left(z^t\right) = \frac{1}{W}\sum_{i=0}^{w-1}\alpha^i \cdot \nabla_z f^{t-i}\left(z^{t-i}\right)$.

**Definition 4.2. Temporal variability:** Many researchers have imposed additional constraints on the variation of the loss function between successive iterations, which is crucial for regret analysis in the online nonconvex case. Drawing on the principle of hyper-regularity, the concept of variation is defined as follows (Jadbabaie et al., 2015; Xu & Zhang, 2024):

$$V^T = \sum_{t=2}^T\left\|f_t(z) - f_{t-1}(z)\right\|, \tag{10}$$

where we denote $\|g(z) - h(z)\| \triangleq \sup_{z \in \mathbb{R}^d}|g(z) - h(z)|$.

## 4.2 ASSUMPTIONS

**Assumption 4.3. Lipschitz gradient of $f^t(z^t)$:** $\nabla_z f^t$ is $L$-Lipschitz continuous, i.e., there exists a constant $L$ for $\forall z_1, z_2 \in \mathbb{R}^d$, such that:

$$\left\|\nabla_z f^t(z_1) - \nabla_z f^t(z_2)\right\|_2 \leq L \left\|z_1 - z_2\right\|_2. \tag{11}$$

**Assumption 4.4. Bounded of $f(z)$:** For all $z \in \mathcal{Z}$, $f_t$ is bounded:

$$|f_t(z)| \leq H. \tag{12}$$

**Assumption 4.5. Bounded of noise:** For all $z \in \mathcal{Z}$, the following inequality is satisfied:

$$|\delta(z)| \leq \Delta. \tag{13}$$

**Assumption 4.6. Bounded of gradient** For all $z \in \mathcal{Z}$, $\hat{\nabla}_z f_\delta^t(z)$ and $\nabla_z f^t(z)$ is bounded:

$$\left\|\hat{\nabla}_z f_\delta^t(z)\right\|_\infty \leq G_\infty, \quad \left\|\nabla_z f^t(z)\right\|_2 \leq G. \tag{14}$$

Assumption 4.3 and 4.4 are the basic assumptions for solving non-convex optimization problems (Ghadimi & Lan, 2013; Hazan & Kale, 2014; Xu et al., 2019; Liu et al., 2020). Assumption 4.5 is a common assumption just to claim the gap between the noisy function $f_\delta(z)$ and the true function $f(z)$, such as random output, different data distributions, and adversarial perturbation (Berahas et al., 2022; Gasnikov et al., 2023; Dvinskikh et al., 2022). In this study, we refer specifically to the noisy output of the generative AI. Assumption 4.6 is critical in non-convex stochastic optimization, as it ensures the fundamental effectiveness of the stochastic gradient (Duchi et al., 2011; Zhou et al., 2018; Chen et al., 2018). Additionally, in experimental settings, it is common practice to impose constraints on the gradients used for updates, such gradient clipping.

## 4.3 LEMMAS

Building on the above assumptions, we further constrain the uncertainty in noisy zeroth-order gradient estimation. Unlike traditional zeroth-order methods (Ghadimi & Lan, 2013; Nesterov & Spokoiny, 2017), Lemma 4.7 and Lemma 4.8 account for the effects of noise in zeroth-order gradient estimation. Specifically, Lemma 4.7 bounds the norm of the estimated gradient, while Lemma 4.8 limits the discrepancy between the estimated and true gradients. This noise stems from the inherent randomness in generative AI outputs, introducing additional variability into the objective function. These lemmas are fundamental to the regret analysis of the subsequent AOZPT algorithm.

**Lemma 4.7. *Bound of the noisy zeroth-order gradient:*** *If $\nabla_z f^t$ is $L$-Lipschitz continuous, and $u^t$ is the direction vector sampled from the unit sphere $\mathcal{S}^d := \{u \in \mathbb{R}^d : \|u\|_2 = 1\}$. Then, the noisy zeroth-order gradient satisfies the following inequality:*

$$\mathbb{E}_{u_t}\left[\left\|\hat{\nabla}_z f_\delta^t(z^t)\right\|_2\right] \leq \frac{L\mu}{2}(d+3)^{\frac{3}{2}} + d\left\|\nabla_z f^t(z^t)\right\|_2 + \frac{\Delta d^{\frac{1}{2}}}{\mu}. \tag{15}$$

**Lemma 4.8. *Bound of the difference between the true gradient and noisy zeroth-order gradient:*** *If $\nabla_z f^t$ is $L$-Lipschitz continuous, and $u^t$ is the direction vector sampled from the unit sphere $\mathcal{S}^d := \{u \in \mathbb{R}^d : \|u\|_2 = 1\}$. Then, the following inequality satisfies:*

$$\left\|\mathbb{E}_{u_t}\left[\hat{\nabla}_z f_\delta^t(z^t)\right] - \nabla_z f_\delta^t(z^t)\right\|_2^2 \leq \frac{2d\Delta^2}{\mu^2} + \frac{L^2\mu^2(d+3)^3}{2}. \tag{16}$$

## 4.4 THE REGRET ANALYSIS FOR AOZPT ALGORITHM

**Theorem 4.9.** *Under Assumption 4.3 - Assumption 4.6, solving the online Black-box prompt learning problem with Algorithm 1. For $t = 1, \ldots, T$, we suppose $\gamma = \frac{\alpha}{\beta^{1/2}} \in (0, 1]$. The following inequality is satisfied:*

$$\mathfrak{R}(T) \leq \mathcal{E}_1 + \mathcal{E}_2 + \mathcal{E}_3. \tag{17}$$

*where*

$$\mathcal{E}_1 = \frac{\left(4H + 2V^T\right)G_\infty}{\eta}, \quad \mathcal{E}_2 = \frac{TG_\infty}{W\epsilon^{\frac{1}{2}}}\left(\frac{2d\Delta^2}{\mu^2} + \frac{L^2\mu^2(d+3)^3}{2}\right),$$

$$\mathcal{E}_3 = \frac{LT\eta M^{\frac{1}{2}}d^{\frac{1}{2}}G_\infty}{2W(1-\gamma)\epsilon^{\frac{1}{2}}}\left(\frac{L\mu(d+3)^{\frac{3}{2}}}{2} + dG + \frac{d^{\frac{1}{2}}\Delta}{\mu}\right).$$

*Futher, we can get:*

$$\mathfrak{R}(T) = \mathcal{O}\left(\frac{T}{W} + \frac{TM^{\frac{1}{2}}}{W}\right). \tag{18}$$

*Remark* 4.10. The $\mathcal{E}_1$ captures the error associated with the standard first-order gradient in regret analysis. The $\mathcal{E}_2$ represents the cumulative zeroth-order variance and generative AI noise encountered during the update process of the AOZPT algorithm, which can be mitigated by adjusting the window length $w$. $\mathcal{E}_3$ is a common term in adaptive algorithms, is similarly influenced by zeroth-order variance and generative AI noise. This highlights the significant impact of these factors on convergence performance. The AOZPT algorithm leverages the adaptive uncertainty scale adjustment to adjust parameters such as $\alpha$, $\beta$ and $w$, effectively limiting their influence. For instance, by setting $\alpha, \beta \to 1^-$ with $\beta \leq \alpha \leq \beta^{\frac{1}{2}}$, and $w = T^{\frac{1}{2}}$, this term can be reduced to a sublinear with respect to $T$. Under these conditions, the AOZPT algorithm can also achieve sublinear regret.

**Proof skeleton of Theorem 4.9:** We begin by establishing the Lipschitz smoothness of the true objective function with respect to the parameters $z$ (Assumption 4.3), a fundamental prerequisite for analyzing the nonconvex optimization problem. In the online nonconvex setting, we further consider the exponentially weighted sliding-window average function to facilitate local regret analysis. Subsequently, we address two primary sources of uncertainty: the variance introduced by zeroth-order gradient and the output noise of the generative AI, as analyzed in Lemmas 4.7 and 4.8. Building upon these assumptions and lemmas, we establish the sublinear regret of the AOZPT algorithm. The detailed proofs of Lemmas 4.7, 4.8, and Theorem 4.9 are provided in Appendix A.2.

## 5 EXPERIMENT

### 5.1 EXPERIMENT SETUP

**Dataset.** We conducted experiments across a range of generative tasks, including text-to-text generation tasks (CNN/DailyMail (Hermann et al., 2015) and GSM8K (Cobbe et al., 2021) datasets) and text-to-image generation tasks (Anime and Painting datasets). For performance evaluation, we selected 500 samples from the CNN/DailyMail and GSM8K datasets, and 150 samples from the Anime and Painting datasets.

**Baselines.** The baselines consist of an online zeroth-order approach and four commonly used classical baselines adapted from an offline setup. The online zeroth-order approach, referred to as "ZO-OGD" for brevity, serves as the primary comparison method, was described in detail by Algorithm 2. For text-to-text generation tasks, the classical baselines include MANUAL PROMPT (MP), In-Context Learning (ICL) (Brown et al., 2020), BDPL (Diao et al., 2022), and RLPROMPT (Deng et al., 2022). For text-to-image generation tasks, the classical baselines are MP, ICL, SFT (Hao et al., 2024), and Promptist (Hao et al., 2024). Additional details regarding the baselines are provided in the Appendix B.

**Evaluation Metrics.** For the text summarization task, the F1-score served as the primary evaluation metric. For the mathematical problem-solving task, accuracy (inverting cumulative binary 0-1 losses) metric was used. For the text-to-image generation task, aesthetic quality was evaluated using the Aesthetic Score Predictor[1], which utilizes CLIP embeddings as input and is trained on the Aesthetic Visual Analysis dataset (Murray et al., 2012).

**Implementation Details.** The experiments are conducted on a machine equipped with a cluster of NVIDIA RTX A6000 GPUs. For text-to-text generation tasks, the open-source model Vicuna-

---

[1]https://github.com/christophschuhmann/improved-aesthetic-predictor

7B[2] was used to generate semantically meaningful prompts. The Llama-3.1-8B[3], GPT-3.5-turbo[4], Qwen2.5-14B[5] and Qwen3-235B [6] models are then employed, with each experiment repeated three times using different random seeds to ensure robustness. For text-to-image generation tasks, the open-source model Vicuna-13B[7] is utilized to produce semantically meaningful prompts. Subsequently, the Dreamlike-photoreal-2.0[8] and Stable Diffusion v1.5[9] models are employed, with each experiment similarly repeated three times using different random seeds for consistency. The implementation code is publicly available at https://github.com/fangjj2000/ICLR_AOZPT_2026.

## 5.2 TEXT-TO-TEXT GENERATION TASKS

We report the average cumulative F1 scores for the text summarization task and the average cumulative accuracy for the mathematical problem task using the Llama-3.1-8B, GPT-3.5-turbo and Qwen2.5-14B models, based on experiments conducted with three random seeds (14, 42, 81). The comparative results for each algorithm across different datasets and models are presented in Table 1. Table 1 demonstrates that AOZPT outperforms four widely used classical algorithms in most cases, highlighting its effectiveness in online settings. Moreover, AOZPT surpasses ZO-OGD, further validating the advantages of its adaptive uncertainty scale adjustment mechanism. In addition, Table A.4 includes the results of the Qwen3-235B model on the GSM8K dataset, which further demonstrate the effectiveness of our method. We also conducted ablation experiments to demonstrate the necessity of open-source LLMs in Table 10.

Table 1: The average cumulative F1 score / accuracy $\pm$ standard deviation using Llama-3.1-8B, GPT-3.5-turbo and Qwen2.5-14B models for CNN/DailyMail, GSM8K Datasets. Each result is reported based on three Monte Carlo experiments. The best results are in bold.

| Dataset | CNN/DailyMail | | | GSM8K | | |
|---|---|---|---|---|---|---|
| Method | Llama-3.1-8B | GPT-3.5-turbo | Qwen2.5-14B | Llama-3.1-8B | GPT-3.5-turbo | Qwen2.5-14B |
| MP | 24.253±0.079 | 34.269±0.035 | 22.068±0.038 | 60.533±0.471 | 69.200±2.209 | 80.200±0.589 |
| ICL | 23.500±0.601 | 32.364±0.259 | 23.064±0.028 | 60.667±0.250 | 69.933±0.806 | 86.733±0.416 |
| BDPL | 23.885±0.280 | 35.372±0.098 | 21.700±3.909 | 37.667±14.055 | 36.406±1.765 | 89.000±0.748 |
| RLPROMPT | 23.618±0.175 | 34.681±0.031 | 20.098±0.579 | 66.867±0.471 | 63.800±2.168 | 81.867±0.094 |
| ZO-OGD | 24.667±0.027 | 34.682±0.291 | 22.034±0.651 | 65.067±5.705 | 69.533±2.532 | 92.533±0.929 |
| AOZPT(Ours) | **24.707±0.047** | **35.399±0.297** | **24.767±0.502** | **69.733±1.514** | **78.133±3.583** | **92.933±0.822** |

Table 2: The average cumulative aesthetic $\pm$ standard deviation using Dreamlike-photoreal-2.0 and Stable Diffusion v1.5 models for Anime, Painting Datasets. Each result is reported based on three Monte Carlo experiments. The best results are in bold.

| Dataset | Anime | | Painting | |
|---|---|---|---|---|
| Method | Dreamlik-2.0 | Stable Diffusion v1.5 | Dreamlike-2.0 | Stable Diffusion v1.5 |
| MP | 5.785±0.002 | 5.336±0.010 | 6.364±0.008 | 5.858±0.011 |
| ICL | 6.133±0.008 | 5.710±0.021 | 6.521±0.016 | 6.074±0.015 |
| SFT | 6.117±0.004 | 5.621±0.025 | 6.645±0.004 | 6.103±0.023 |
| Promptist | 6.093±0.010 | 5.579±0.006 | 6.552±0.004 | 6.011±0.022 |
| ZO-OGD | 6.263±0.024 | 5.892±0.039 | 6.602±0.053 | 6.287±0.013 |
| AOZPT (Ours) | **6.282±0.021** | **5.930±0.015** | **6.656±0.015** | **6.313±0.009** |

## 5.3 TEXT-TO-IMAGE GENERATION TASKS

We present the average cumulative aesthetic score for the Dreamlike-photoreal-2.0 and Stable Diffusion v1.5 models on the text-to-image generation tasks (Anime and Painting datasets), based

[2]https://huggingface.co/lmsys/vicuna-7b-v1.5

[3]https://huggingface.co/meta-llama/Llama-3.1-8B-Instruct

[4]https://openai.com/index/openai-api/

[5]https://huggingface.co/Qwen/Qwen2.5-14B-Instruct-1M

[6]https://huggingface.co/Qwen/Qwen3-235B-A22B-Instruct-2507

[7]https://huggingface.co/lmsys/vicuna-13b-v1.3

[8]https://huggingface.co/dreamlike-art/dreamlike-photoreal-2.0

[9]https://huggingface.co/stable-diffusion-v1-5/stable-diffusion-v1-5

on experiments conducted with three random seeds. The comparative results for each algorithm across different datasets and models are provided in Table 2. Table 2 shows that AOZPT outperforms baseline methods in most cases, demonstrating its effectiveness in online scenarios. Table 6 presents text-to-image experiments conducted under data drift conditions. The results indicate that under varying levels of data drift (10, 50, 75, 150), the online black-box optimization algorithms achieve higher aesthetic. In Table 7, ablation experiments are also included to illustrate the role of the adaptive uncertainty scale adjustment mechanism and the online zero-order gradient method in prompt optimization. Additionally, we provide a performance comparison between our adaptive uncertainty scale adjustment mechanism and several widely-used adaptive gradient algorithms in Table 12. Lastly, we present a case study of the Anime and Painting dataset in Table 16 and Table 17.

# 6 DISCUSSION

## 6.1 ONLINE LEARNING VS. OFFLINE LEARNING

Offline learning offers distinct advantages, particularly for fixed-task datasets, by enabling stable training processes and achieving high accuracy. However, its primary application lies in developing deployable model products, as it lacks adaptability to evolving data. When confronted with dynamic data streams, offline learning requires retraining the model on the entire dataset, encompassing both historical and newly acquired data. This process incurs substantial computational costs and training inefficiencies. Each time the data changes, the model must be retrained from scratch, rendering this approach unsuitable for scenarios demanding real-time responses and frequent updates. This limitation stems not from the model itself but from the offline learning paradigm. By contrast, online learning provides a more effective alternative. It incrementally processes streaming data and updates the model in real time, enabling dynamic adaptation to data changes. Rather than retraining the entire model, online learning continuously updates and optimizes it based on current inputs, thereby reducing computational overhead and enhancing responsiveness.

## 6.2 REAL-WORLD, NON-HYPOTHETICAL APPLICATION SCENARIOS

**Emotion-responsive chatbots and intelligent tutoring systems**—are not hypothetical constructs, but are grounded in real product requirements. These systems must adapt their response styles and content in real time based on user feedback, rendering long-cycle model fine-tuning or manual prompt redesign impractical. Consequently, online black-box prompt optimization presents a broadly applicable solution for real-world deployment. For instance, emotion-aware chatbots such as Replika and Woebot adjust their tone in response to users' emotional states, despite lacking access to internal model weights. Similarly, language learning platforms like Duolingo Max and Socratic dynamically tailor instructional content and tone based on student performance. In both cases, real-time model adaptation is infeasible, necessitating input-side prompt adjustments to enable responsive and personalized behavior. To further illustrate the practical applicability of our method, we present additional examples from **high-stakes domains such as healthcare, finance, and law** in Appendix C, where the feature distribution of input data is rarely stationary.

# 7 CONCLUSION

In this paper, we propose AOZPT, a novel approach that combines black-box prompt tuning with online learning. This method utilizes a frozen open-source LLM for instance optimization, leveraging the LLM's advanced understanding of language patterns and context to optimize soft prompts for generating high-quality, semantically rich prompts. During the prompt updating phase, AOZPT dynamically adjusts prompts for generative AI based on streaming data, eliminating the need for retraining on the entire dataset. To address the variance in zeroth-order gradient estimation and the noise in generative AI, we introduce an adaptive uncertainty scaling mechanism. This mechanism incorporates the exponentially weighted moving average of gradients into the update process, effectively reducing gradient uncertainty. To validate the effectiveness of AOZPT, we performed a formal regret analysis in non-convex settings, demonstrating that sublinear regret convergence is achievable. Furthermore, we evaluated AOZPT on both text-to-text (CNN/DailyMail and GSM8K) and text-to-image (Anime and Painting) tasks in simulated online scenarios, with results consistently indicating that AOZPT outperforms baseline models.

## ACKNOWLEDGMENTS

Dr. Yi Chang was supported by the National Key R&D Program of China (Grant No. 2023YFF0905400), the National Natural Science Foundation of China (Grant No. U2341229), the Fundamental and Interdisciplinary Disciplines Breakthrough Plan of the Ministry of Education of China (JYB2025XDXM903), and the New Cornerstone Science Foundation through the XPLORER PRIZE.

## ETHICS STATEMENT

All participants in this work, as well as the paper submission, adhere to the ICLR Code of Ethics ( https://iclr.cc/public/CodeOfEthics).

## REPRODUCIBILITY STATEMENT

We affirm that the results of this work are fully reproducible. Appendix A.2 provides the theoretical proofs. Appendix B details the experimental implementations, and the source code will be publicly released after publication of the paper.

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

# A CONVERGENCE ANALYSIS

## A.1 NOTATION

Table 3: Notations.

| Symbolic | Meaning |
|---|---|
| $t = 1, ..., T$ | Number of iterations |
| $\|\cdot\|_p$ | p-norm |
| $\delta$ | Noise of model's output |
| $\mathbb{E}$ | Expectation |
| $\xi = \{x, y\}$ | Sample |
| $\mathcal{F}$ | Open-source LLM |
| $A \in \mathbb{R}^{D \times d}$ | Random projection matrix |
| $z^t \in \mathbb{R}^d$ | Optimized low-dimensional vector |
| $\phi_0$ | The initial prompt |
| $\phi_t \in \Phi$ | The discrete prompt |
| $\mathcal{G}$ | Generative model |
| $\ell$ | Loss function |
| $f$ | Objective function |
| $F$ | Sliding-window average function |
| $w$ | Length of window |
| $\alpha$ | Weight |
| $\nabla_z f$ | The full gradient |
| $\hat{\nabla}_z f$ | The zeroth-order gradient |
| $\hat{\nabla}_z f_\delta$ | The noisy zeroth-order gradient |
| $g$ | $\hat{\nabla}_z f_\delta$ |
| $\mathbf{V}$ | $diag(v + \epsilon)$ |

## A.2 PROOFS OF IMPORTANT LEMMAS AND THEOREMS

Proof of Lemma 4.7:

*Proof.* According to the definition (4):

$$
\mathbb{E}_u \left[ \left\| \hat{\nabla}_z f_\delta^t \left( z^t \right) \right\|_2 \right]
$$

$$
= \mathbb{E}_u \left[ \left\| \frac{f_\delta^t \left( z^t + \mu u^t \right) - f_\delta^t \left( z^t - \mu u^t \right)}{2\mu} u^t \right\|_2 \right]
$$

$$
= \frac{1}{2\mu} \mathbb{E}_u \left[ \left\| \left( f_\delta^t \left( z^t + \mu u^t \right) - f_\delta^t \left( z^t - \mu u^t \right) \right) u^t \right\|_2 \right]
$$

$$
\overset{(1)}{\leq} \frac{1}{2\mu} \underbrace{\mathbb{E}_u \left[ \left\| \left( f^t \left( z^t + \mu u^t \right) - f^t \left( z^t - \mu u^t \right) \right) u^t \right\|_2 \right]}_{a)} + \frac{1}{2\mu} \underbrace{\mathbb{E}_u \left[ \left\| \left( \delta \left( z^t + \mu u^t \right) - \delta \left( z^t - \mu u^t \right) \right) u^t \right\|_2 \right]}_{b)},
$$

$$(19)$$

where (1) use the inequality $\|a + b\|_2 \leq \|a\|_2 + \|b\|_2$ and definition (3).

Then, for a):

$$
\mathbb{E}_u \left\| \left( f^t \left( z^t + \mu u^t \right) - f^t \left( z^t - \mu u^t \right) \right) u^t \right\|_2
$$

$$
\overset{(1)}{\leq} \mathbb{E}_u \left\| \left( f^t \left( z^t + \mu u^t \right) - f^t \left( z^t \right) - \left\langle \nabla_z f^t \left( z^t \right), \mu u^t \right\rangle \right) u^t \right\|_2
$$

$$
+ \mathbb{E}_u \left\| \left( f^t \left( z^t - \mu u^t \right) - f^t \left( z^t \right) + \left\langle \nabla_z f^t \left( z^t \right), \mu u^t \right\rangle \right) u^t \right\|_2 + \mathbb{E}_u \left\| 2 \left\langle \nabla_z f^t \left( z^t \right), \mu u^t \right\rangle u^t \right\|_2
$$

$$
\overset{(2)}{\leq} 2\mathbb{E}_u \left\| \frac{L}{2} \mu^2 \left\| u^t \right\|_2^2 u^t \right\|_2 + 2\mathbb{E}_u \left\| \left\langle \nabla_z f^t \left( z^t \right), \mu u^t \right\rangle u^t \right\|_2
$$

$$
= L\mu^2 \mathbb{E}_u \left\| u^t \right\|_2^3 + 2\mu \mathbb{E}_u \left\| \left\langle \nabla_z f^t \left( z^t \right), u^t \right\rangle \right\|_2 \left\| u^t \right\|_2
$$

$$
\overset{(3)}{\leq} L\mu^2 (d+3)^{\frac{3}{2}} + 2\mu d \left\| \nabla_z f^t \left( z^t \right) \right\|_2, \tag{20}
$$

where (1) use inequality $\|a + b + c\|_2 \leq \|a\|_2 + \|b\|_2 + \|c\|_2$; (2) uses the Assumption 4.3; (3) use the Lemma 1 in Nesterov & Spokoiny (2017).

For (b), using Assumption 4.5 and the inequalities (A.1)–(A.2), we have

$$
\mathbb{E}_{u,\delta} \left[ \left\| \left( \delta \left( z^t + \mu u^t \right) - \delta \left( z^t - \mu u^t \right) \right) u^t \right\|_2 \right]
$$

$$
\leq \left( \mathbb{E}_{u,\delta} \left[ \left( \delta \left( z^t + \mu u^t \right) - \delta \left( z^t - \mu u^t \right) \right)^2 \right] \right)^{\frac{1}{2}} \left( \mathbb{E}_u \left[ \left\| u^t \right\|_2^2 \right] \right)^{\frac{1}{2}}
$$

$$
\overset{(1)}{\leq} 2\sigma d^{\frac{1}{2}},
$$

where (1) follows from (A.2), and from Lemma 1 in Nesterov & Spokoiny (2017).

Finally, we take a) and b) into (19):

$$
\mathbb{E}_u \left[ \left\| \hat{\nabla}_z f_\delta^t \left( z^t \right) \right\|_2^2 \right]
$$

$$
\leq \frac{1}{2\mu} \cdot \left( L\mu^2 (d+3)^{\frac{3}{2}} + 2\mu d \left\| \nabla_z f^t \left( z^t \right) \right\|_2 \right) + \frac{1}{2\mu} \cdot 2\sigma d^{\frac{1}{2}}
$$

$$
= \frac{L\mu}{2} (d+3)^{\frac{3}{2}} + d \left\| \nabla_z f^t \left( z^t \right) \right\|_2 + \frac{\sigma d^{\frac{1}{2}}}{\mu}.
$$

$\square$

Proof of Lemma 4.8:

*Proof.*

$$
\left\| \mathbb{E}_u \left[ \hat{\nabla}_z f_\delta^t \left( z^t \right) \right] - \nabla_z f^t \left( z^t \right) \right\|_2^2
$$

$$
\leq 2 \left\| \mathbb{E}_u \left[ \hat{\nabla}_z f_\delta^t \left( z^t \right) \right] - \mathbb{E}_u \left[ \hat{\nabla}_z f^t \left( z^t \right) \right] \right\|_2^2 + 2 \left\| \mathbb{E}_u \left[ \hat{\nabla}_z f^t \left( z^t \right) \right] - \nabla_z f^t \left( z^t \right) \right\|_2^2
$$

$$
\overset{(1)}{\leq} 2 \mathbb{E}_u \left[ \left\| \frac{\delta(z^t + \mu u^t) - \delta(z^t - \mu u^t)}{2\mu} u^t \right\|_2^2 \right] + \frac{L^2 \mu^2 (d+3)^3}{2}
$$

$$
\overset{(2)}{\leq} \frac{2\sigma^2 d}{\mu^2} + \frac{L^2 \mu^2 (d+3)^3}{2}.
$$

$\square$

where (1) uses Jensen's inequality and Lemma 3 in Nesterov & Spokoiny (2017); (2) uses inequalities (A.1)–(A.2) derived from Assumption 4.5.

**Lemma A.1.** *For $t = 1, \ldots, T$, $\alpha, \beta$ are the weight parameters, and $\gamma = \alpha/\beta^{1/2}$. To simplify the expression, we denote $\hat{\nabla}_z f_\delta^t(z^t)$ as $\mathbf{g}^t$. And we denote $\mathbf{V}_t = v_t + \epsilon$. Suppose that $\gamma \leq 1$, then we have the following inequality for :*

$$\sum_{t=1}^{T} \left\| \mathbf{V}_t^{-\frac{1}{2}} \mathbf{m}_t \right\|_2^2 \leq \frac{d^{1/2} M^{\frac{1}{2}}}{2W(1-\gamma)\epsilon^{\frac{1}{2}}} \sum_{t=1}^{T} \left\| g^t \right\|_2. \tag{21}$$

Proof of Lemma A.1:

*Proof.* Recall that $v_{t,j}, m_{t,j}, g_{t,j}$ denote the $j$-th coordinate of $\mathbf{v}_t, \mathbf{m}_t$ and $\mathbf{g}^t$. We have

$$
\begin{aligned}
\| \mathbf{V}_t^{-\frac{1}{2}} \mathbf{m}_t \|_2^2 &= \sum_{j=1}^{d} \frac{m_{t,j}^2}{v_{t,j}^{\frac{1}{2}}} \cdot \frac{v_{t,j}^{\frac{1}{2}}}{v_{t,j} + \epsilon} \\
&\overset{(1)}{\leq} \sum_{j=1}^{d} \frac{m_{t,j}^2}{v_{t,j}^{\frac{1}{2}}} \cdot \frac{v_{t,j}^{\frac{1}{2}}}{2 v_{t,j}^{\frac{1}{2}} \epsilon^{\frac{1}{2}}} \\
&= \frac{1}{2\epsilon^{\frac{1}{2}}} \sum_{j=1}^{d} \frac{m_{t,j}^2}{v_{t,j}^{\frac{1}{2}}} \\
&= \frac{M^{\frac{1}{2}}}{2W^2 \epsilon^{\frac{1}{2}}} \sum_{j=1}^{d} \frac{\left( \sum_{i=0}^{w-1} \alpha^i g_{t-i,j} \right)^2}{\left( \sum_{i=0}^{w-1} \beta^i g_{t-i,j}^2 \right)^{\frac{1}{2}}},
\end{aligned}
\tag{22}
$$

where (1) is use inequality $a + b \geq 2\sqrt{ab}$ . Next we have

$$
\begin{aligned}
\frac{M^{\frac{1}{2}} \eta^2}{2W^2 \epsilon^{\frac{1}{2}}} \sum_{j=1}^{d} \frac{\left( \sum_{i=0}^{w-1} \alpha^i g_{t-i,j} \right)^2}{\left( \sum_{i=0}^{w-1} \beta^i g_{t-i,j}^2 \right)^{\frac{1}{2}}} &\leq \frac{M^{\frac{1}{2}} \eta^2}{2W^2 \epsilon^{\frac{1}{2}}} \sum_{j=1}^{d} \frac{\left( \sum_{i=0}^{w-1} \alpha^i \right) \left( \sum_{i=0}^{w-1} \alpha^i |g_{t-i,j}|^2 \right)}{\left( \sum_{i=0}^{w-1} \beta^i g_{t-i,j}^2 \right)^{\frac{1}{2}}} \\
&= \frac{M^{\frac{1}{2}} \eta^2}{2W \epsilon^{\frac{1}{2}}} \sum_{j=1}^{d} \frac{\sum_{i=0}^{w-1} \alpha^i |g_{t-i,j}|^2}{\left( \sum_{i=0}^{w-1} \beta^i g_{t-i,j}^2 \right)^{\frac{1}{2}}} \\
&\leq \frac{M^{\frac{1}{2}} \eta^2}{2W \epsilon^{\frac{1}{2}}} \sum_{j=1}^{d} \sum_{i=0}^{w-1} \frac{\alpha^i |g_{t-i,j}|^2}{\left( \beta^i g_{t-i,j}^2 \right)^{\frac{1}{2}}} \\
&= \frac{M^{\frac{1}{2}} \eta^2}{2W \epsilon^{\frac{1}{2}}} \sum_{j=1}^{d} \sum_{i=0}^{w-1} (\frac{\alpha}{\beta^{\frac{1}{2}}})^i |g_{t-i,j}| \\
&= \frac{M^{\frac{1}{2}} \eta^2}{2W \epsilon^{\frac{1}{2}}} \sum_{j=1}^{d} \sum_{i=0}^{w-1} \gamma^i |g_{t-i,j}|,
\end{aligned}
\tag{23}
$$

where the first inequality holds due to Cauchy inequality. The last equality holds due to the definition of $\gamma$. Telescoping (23) for $t = 1$ to $T$, we have:

$$
\begin{aligned}
\sum_{t=1}^{T} \|\mathbf{V}_t^{-\frac{1}{2}} \mathbf{m}_t\|_2^2 &\leq \frac{M^{\frac{1}{2}}}{2W\epsilon^{\frac{1}{2}}} \sum_{t=1}^{T} \sum_{j=1}^{d} \sum_{i=0}^{w-1} \gamma^i |g_{t-i,j}| \\
&\overset{(1)}{\leq} \frac{M^{\frac{1}{2}}}{2W\epsilon^{\frac{1}{2}}} \sum_{t=1}^{T} \sum_{j=1}^{d} \sum_{i=0}^{t} \gamma^i |g_{t-i,j}| \\
&= \frac{M^{\frac{1}{2}}}{2W\epsilon^{\frac{1}{2}}} \sum_{j=1}^{d} \sum_{t=1}^{T} |g_{t,j}| \sum_{i=0}^{t} \gamma^i \\
&\overset{(2)}{\leq} \frac{M^{\frac{1}{2}}}{2W(1-\gamma)\epsilon^{\frac{1}{2}}} \sum_{t=1}^{T} \sum_{j=1}^{d} |g_{t,j}| \\
&\overset{(3)}{\leq} \frac{M^{\frac{1}{2}}}{2W(1-\gamma)\epsilon^{\frac{1}{2}}} \sum_{t=1}^{T} \left( \sum_{j=1}^{d} g_{t,j}^2 \right)^{1/2} \cdot d^{1/2} \\
&= \frac{d^{1/2} M^{\frac{1}{2}}}{2W(1-\gamma)\epsilon^{\frac{1}{2}}} \sum_{t=1}^{T} \|g^t\|_2 ,
\end{aligned}
\tag{24}
$$

where (1) is because $f_\delta^t(z^t) = 0$ for $t \leq 0$; (2) is because $\gamma \leq 1$; (3) holds due to Hölder's inequality. $\qquad \square$

Proof Theorem 4.9:

*Proof.* Since $\nabla_z f^t(z^t)$ is $L$-Lipschitz continuous, $\nabla_z F_{w,\alpha}^t(z^t)$ is also $L$-Lipschitz continuous, then we can get:

$$
\begin{aligned}
&F_{w,\alpha}^t(z^{t+1}) - F_{w,\alpha}^t(z^t) \\
&\leq \langle \nabla_z F_{w,\alpha}^t(z^t), z^{t+1} - z^t \rangle + \frac{L}{2} \|z^{t+1} - z^t\|_2^2.
\end{aligned}
\tag{25}
$$

We take the expectation about $\{u^r\}_{r=t-w+1}^t$ for both sides, then we simplify $\mathbb{E}_{\{u^r\}_{r=t-w+1}^t}$ to $\mathbb{E}_u$:

$$
\underbrace{\mathbb{E}_u \left[ F_{w,\alpha}^t(z^{t+1}) - F_{w,\alpha}^t(z^t) \right]}_{a)}
$$
$$
\leq \underbrace{\mathbb{E}_u \left[ \langle \nabla_z F_{w,\alpha}^t(z^t), z^{t+1} - z^t \rangle \right]}_{b)} + \frac{L}{2} \mathbb{E}_u \left[ \|z^{t+1} - z^t\|_2^2 \right].
\tag{26}
$$

For a):

$$
\begin{aligned}
&\mathbb{E}_u \left[ F_{w,\alpha}^t(z^{t+1}) - F_{w,\alpha}^t(z^t) \right] \\
&= F_{w,\alpha}^t(z^{t+1}) - F_{w,\alpha}^t(z^t) \\
&= F_{w,\alpha}^t(z^{t+1}) - F_{w,\alpha}^{t+1}(z^{t+1}) + F_{w,\alpha}^{t+1}(z^{t+1}) - F_{w,\alpha}^t(z^t).
\end{aligned}
\tag{27}
$$

For b):

$$
\begin{aligned}
&\mathbb{E}_u \left[ \left\langle \nabla_z F_{w,\alpha}^t \left( z^t \right), z^{t+1} - z^t \right\rangle \right] \\
&= \left\langle \nabla_z F_{w,\alpha}^t \left( z^t \right), -\eta \cdot \mathbb{E}_u \left[ \mathbf{V}_t^{-\frac{1}{2}} \hat{\nabla}_z F_{\delta,w,\alpha}^t \left( z^t \right) \right] \right\rangle \\
&= \mathbb{E}_u \left[ \left\langle \nabla_z F_{w,\alpha}^t \left( z^t \right), -\eta \cdot \mathbf{V}_t^{-\frac{1}{2}} \nabla_z F_{w,\alpha}^t \left( z^t \right) \right\rangle \right] \\
&\quad + \mathbb{E}_u \left[ \left\langle \nabla_z F_{w,\alpha}^t \left( z^t \right), \eta \cdot \mathbf{V}_t^{-\frac{1}{2}} \left( \nabla_z F_{w,\alpha}^t \left( z^t \right) - \mathbf{m}_t \right) \right\rangle \right] \\
&= \mathbb{E}_u \left[ \left\langle \nabla_z F_{w,\alpha}^t \left( z^t \right), -\eta \cdot \mathbf{V}_t^{-\frac{1}{2}} \nabla_z F_{w,\alpha}^t \left( z^t \right) \right\rangle \right] \\
&\quad + \eta \cdot \mathbb{E}_u \left[ \left\langle \nabla_z F_{w,\alpha}^t \left( z^t \right) \cdot \mathbf{V}_t^{-\frac{1}{4}}, \mathbf{V}_t^{-\frac{1}{4}} \left( \nabla_z F_{w,\alpha}^t \left( z^t \right) - \mathbf{m}_t \right) \right\rangle \right] \\
&\overset{(1)}{\leq} \mathbb{E}_u \left[ \left\langle \nabla_z F_{w,\alpha}^t \left( z^t \right), -\eta \cdot \mathbf{V}_t^{-\frac{1}{2}} \nabla_z F_{w,\alpha}^t \left( z^t \right) \right\rangle \right] + \mathbb{E}_u \left[ \left\langle \nabla_z F_{w,\alpha}^t \left( z^t \right), \frac{\eta}{2} \cdot \mathbf{V}_t^{-\frac{1}{2}} \nabla_z F_{w,\alpha}^t \left( z^t \right) \right\rangle \right] \\
&\quad + \frac{\eta}{2} \mathbb{E}_u \left[ \left\| \left( \nabla_z F_{w,\alpha}^t \left( z^t \right) - \mathbf{m}_t \right) \cdot \mathbf{V}_t^{-\frac{1}{4}} \right\|_2^2 \right] \\
&= \mathbb{E}_u \left[ \left\langle \nabla_z F_{w,\alpha}^t \left( z^t \right), -\frac{\eta}{2} \cdot \mathbf{V}_t^{-\frac{1}{2}} \nabla_z F_{w,\alpha}^t \left( z^t \right) \right\rangle \right] + \frac{\eta}{2} \mathbb{E}_u \left[ \left\| \left( \nabla_z F_{w,\alpha}^t \left( z^t \right) - \mathbf{m}_t \right) \cdot \mathbf{V}_t^{-\frac{1}{4}} \right\|_2^2 \right],
\end{aligned}
\tag{28}
$$

where (1) is use inequality $\langle a, b \rangle \leq \frac{\|a\|_2^2 + \|b\|_2^2}{2}$.

Organizing the (27) and (28) into (26), we can get:

$$
\begin{aligned}
&F_{w,\alpha}^t \left( z^{t+1} \right) - F_{w,\alpha}^{t+1} \left( z^{t+1} \right) + F_{w,\alpha}^{t+1} \left( z^{t+1} \right) - F_{w,\alpha}^t \left( z^t \right) \\
&\leq \mathbb{E}_u \left[ \left\langle \nabla_z F_{w,\alpha}^t \left( z^t \right), -\frac{\eta}{2} \cdot \mathbf{V}_t^{-\frac{1}{2}} \nabla_z F_{w,\alpha}^t \left( z^t \right) \right\rangle \right] \\
&\quad + \frac{\eta}{2} \mathbb{E}_u \left[ \left\| \left( \nabla_z F_{w,\alpha}^t \left( z^t \right) - \mathbf{m}_t \right) \cdot \mathbf{V}_t^{-\frac{1}{4}} \right\|_2^2 \right] + \frac{L\eta^2}{2} \mathbb{E}_u \left[ \left\| \mathbf{V}_t^{-\frac{1}{2}} \mathbf{m}_t \right\|_2^2 \right].
\end{aligned}
\tag{29}
$$

Because of $\mathbf{V}_t \leq G_\infty^2 \mathbf{I}$, refer to Lemma 6.4 in Zhou et al. (2018), we get:

$$
\begin{aligned}
&F_{w,\alpha}^t \left( z^{t+1} \right) - F_{w,\alpha}^{t+1} \left( z^{t+1} \right) + F_{w,\alpha}^{t+1} \left( z^{t+1} \right) - F_{w,\alpha}^t \left( z^t \right) \\
&\leq -\frac{\eta}{2G_\infty} \mathbb{E}_u \left[ \left\| \nabla_z F_{w,\alpha}^t \left( z^t \right) \right\|_2^2 \right] + \frac{\eta}{2} \mathbb{E}_u \left[ \left\| \left( \nabla_z F_{w,\alpha}^t \left( z^t \right) - \mathbf{m}_t \right) \cdot \mathbf{V}_t^{-\frac{1}{4}} \right\|_2^2 \right] \\
&\quad + \frac{L\eta^2}{2} \mathbb{E}_u \left[ \left\| \mathbf{V}_t^{-\frac{1}{2}} \mathbf{m}_t \right\|_2^2 \right].
\end{aligned}
\tag{30}
$$

and for both sides take $t = 1, ..., T$ in (30) gives:

$$
\begin{aligned}
\frac{\eta}{2G_\infty} \sum_{t=1}^T \mathbb{E}_u \left[ \nabla_z F_{w,\alpha}^t \left( z^t \right) \right] &\leq \underbrace{\sum_{t=1}^T F_{w,\alpha}^t \left( z^{t+1} \right) - F_{w,\alpha}^{t+1} \left( z^{t+1} \right) + F_{w,\alpha}^{t+1} \left( z^{t+1} \right) - F_{w,\alpha}^t \left( z^t \right)}_{a)} \\
&\quad + \underbrace{\frac{\eta}{2} \sum_{t=1}^T \mathbb{E}_u \left[ \left\| \left( \nabla_z F_{w,\alpha}^t \left( z^t \right) - \mathbf{m}_t \right) \cdot \mathbf{V}_t^{-\frac{1}{4}} \right\|_2^2 \right]}_{b)} \\
&\quad + \underbrace{\frac{L\eta^2}{2} \sum_{t=1}^T \mathbb{E}_u \left[ \left\| \mathbf{V}_t^{-\frac{1}{2}} \mathbf{m}_t \right\|_2^2 \right]}_{c)}.
\end{aligned}
\tag{31}
$$

For (a):

$$\sum_{t=1}^{T} F_{w,\alpha}^{t} \left( z^{t+1} \right) - F_{w,\alpha}^{t+1} \left( z^{t+1} \right) + F_{w,\alpha}^{t+1} \left( z^{t+1} \right) - F_{w,\alpha}^{t} \left( z^{t} \right)$$

$$= \sum_{t=1}^{T} \left( F_{w,\alpha}^{t} \left( z^{t} \right) - F_{w,\alpha}^{t+1} \left( z^{t+1} \right) \right) + \sum_{t=1}^{T} \left( F_{w,\alpha}^{t+1} \left( z^{t+1} \right) - F_{w,\alpha}^{t} \left( z^{t+1} \right) \right)$$

$$= \frac{1}{W} \sum_{i=0}^{w-1} \alpha^{i} \cdot \sum_{t=1}^{T} \left( f^{t-i} \left( z^{t-i} \right) - f^{t+1-i} \left( z^{t+1-i} \right) \right)$$

$$+ \frac{1}{W} \sum_{i=0}^{w-1} \alpha^{i} \cdot \sum_{t=1}^{T} \left( f^{t+1-i} \left( z^{t+1-i} \right) - f^{t-i} \left( z^{t+1-i} \right) \right)$$

$$\overset{(1)}{\leq} 2H + V^{T}. \tag{32}$$

where (1) use Assumption 4.4, Definition (10).

For (b):

$$\frac{\eta}{2} \sum_{t=1}^{T} \mathbb{E}_{u} \left[ \left\| \left( \nabla_{z} F_{w,\alpha}^{t} \left( z^{t} \right) - \mathbf{m}_{t} \right) \cdot \mathbf{V}_{t}^{-\frac{1}{4}} \right\|_{2}^{2} \right]$$

$$\overset{(1)}{\leq} \frac{\eta}{2\epsilon^{\frac{1}{2}}} \sum_{t=1}^{T} \mathbb{E}_{u} \left\| \nabla_{z} F_{w,\alpha}^{t} \left( z^{t} \right) - \hat{\nabla}_{z} F_{\delta,w,\alpha}^{t} \left( z^{t} \right) \right\|_{2}^{2}$$

$$= \frac{\eta}{2\epsilon^{\frac{1}{2}}} \sum_{t=1}^{T} \mathbb{E}_{u} \left\| \frac{1}{W} \sum_{i=0}^{w-1} \alpha^{i} \cdot \left[ \nabla_{z} f^{t-i} \left( z^{t-i} \right) - \hat{\nabla}_{z} f_{\delta}^{t-i} \left( z^{t-i} \right) \right] \right\|_{2}^{2}$$

$$\overset{(2)}{=} \frac{\eta}{2W^{2}\epsilon^{\frac{1}{2}}} \sum_{t=1}^{T} \sum_{i=0}^{w-1} (\alpha^{i})^{2} \cdot \mathbb{E}_{u^{t-i}} \left\| \nabla_{z} f^{t-i} \left( z^{t-i} \right) - \hat{\nabla}_{z} f_{\delta}^{t-i} \left( z^{t-i} \right) \right\|_{2}^{2}$$

$$\overset{(3)}{\leq} \frac{\eta T}{2W\epsilon^{\frac{1}{2}}} \left( \frac{2d\sigma^{2}}{\mu^{2}} + \frac{L^{2}\mu^{2}(d+3)^{3}}{2} \right). \tag{33}$$

where (1) is because $\mathbf{V}_{t} \geq \epsilon \mathbf{I}$; (2) is because the sampling of $u^{t-i}$ is independent; (3) uses $W = \sum_{i=0}^{w-1} \alpha^{i}$, $0 < \alpha < 1$ and Lemma 4.8.

Finally, we can get:

$$\frac{\eta}{2} \sum_{t=1}^{T} \mathbb{E}_{u} \left[ \left\| \nabla_{z} F_{w,\alpha}^{t} \left( z^{t} \right) - m_{t} \right\|_{2}^{2} \right] \leq \frac{\eta T}{2W\epsilon^{\frac{1}{2}}} \left( \frac{2d\sigma^{2}}{\mu^{2}} + \frac{L^{2}\mu^{2}(d+3)^{3}}{2} \right). \tag{34}$$

For (c):

$$\frac{L\eta^{2}}{2} \sum_{t=1}^{T} \mathbb{E}_{u} \left[ \left\| \mathbf{V}_{t}^{-\frac{1}{2}} \mathbf{m}_{t} \right\|_{2}^{2} \right]$$

$$\overset{(1)}{\leq} \frac{L\eta^{2}}{2} \frac{d^{1/2} M^{\frac{1}{2}}}{2W(1-\gamma)\epsilon^{\frac{1}{2}}} \sum_{t=1}^{T} \mathbb{E}_{u} \left[ \left\| g^{t} \right\|_{2} \right]$$

$$\overset{(2)}{\leq} \frac{LT\eta^{2} d^{\frac{1}{2}} M^{\frac{1}{2}}}{4W(1-\gamma)\epsilon^{\frac{1}{2}}} \left( \frac{L\mu}{2} (d+3)^{\frac{3}{2}} + d \left\| \nabla_{z} f^{t} \left( z^{t} \right) \right\|_{2} + \frac{\sigma d^{\frac{1}{2}}}{\mu} \right)$$

$$\overset{(3)}{\leq} \frac{LT\eta^{2} d^{\frac{1}{2}} M^{\frac{1}{2}}}{4W(1-\gamma)\epsilon^{\frac{1}{2}}} \left( \frac{L\mu(d+3)^{\frac{3}{2}}}{2} + dG + \frac{d^{\frac{1}{2}}\sigma}{\mu} \right), \tag{35}$$

where (1) uses Lemma A.1 ; (2) uses Lemma 4.7; (3) uses Assumption 4.6.
We take a), b) and c) into (31):

$$\frac{\eta}{2G_\infty} \sum_{t=1}^{T} \mathbb{E}_u \left[ \left\| \nabla_z F_{w,\alpha}^t \left( z^t \right) \right\|_2^2 \right]$$

$$\leq 2H + V^T + \frac{\eta T}{2W\epsilon^{\frac{1}{2}}} \left( \frac{2d\sigma^2}{\mu^2} + \frac{L^2\mu^2(d+3)^3}{2} \right) + \frac{LT\eta^2 M^{\frac{1}{2}} d^{\frac{1}{2}}}{2W(1-\gamma)\epsilon^{\frac{1}{2}}} \left( \frac{L\mu(d+3)^{\frac{3}{2}}}{2} + dG + \frac{d^{\frac{1}{2}}\sigma}{\mu} \right).$$

Divide both sides simultaneously by $\frac{\eta}{2G_\infty}$:

$$\sum_{t=1}^{T} \left[ \left\| \nabla_z F_{w,\alpha}^t \left( z^t \right) \right\|_2^2 \right]$$

$$\leq \frac{\left( 4H + 2V^T \right) G_\infty}{\eta} + \frac{T G_\infty}{W\epsilon^{\frac{1}{2}}} \left( \frac{2d\sigma^2}{\mu^2} + \frac{L^2\mu^2(d+3)^3}{2} \right)$$

$$+ \frac{LT\eta M^{\frac{1}{2}} d^{\frac{1}{2}} G_\infty}{2W(1-\gamma)\epsilon^{\frac{1}{2}}} \left( \frac{L\mu(d+3)^{\frac{3}{2}}}{2} + dG + \frac{d^{\frac{1}{2}}\sigma}{\mu} \right).$$

$\square$

## A.3 Theoretical Analysis under Sub-Gaussian Noise

**Sub-Gaussian noise:** At each round $t$, the learner observes a noisy function value

$$\tilde{f}_\delta^t(z^t) = f^t(z^t) + \delta(z^t),$$

where $\delta(z^t)$ denotes the observation noise at round $t$. We assume that $\{\delta(z^t)\}_{t\geq 1}$ forms a martingale difference sequence and is uniformly $\sigma$-sub-Gaussian: there exists a constant $\sigma > 0$ such that for all $t \geq 1$ and all $\lambda \in \mathbb{R}$,

$$\mathbb{E}\big[\delta(z^t) \mid \mathcal{F}_{t-1}\big] = 0, \qquad \mathbb{E}\Big[ \exp\big(\lambda\,\delta(z^t)\big) \Big| \mathcal{F}_{t-1}\Big] \leq \exp\Big(\frac{\lambda^2\sigma^2}{2}\Big),$$

where $\{\mathcal{F}_t\}$ denotes the filtration generated by all randomness and observations up to round $t - 1$.

Although this modification alters the statements and proofs of Lemma 4.7 and Lemma 4.8 and affects the final regret bounds, it does not compromise the convergence guarantees of the proposed algorithm. We now proceed to provide a detailed analysis of the specific adjustments required in the theoretical derivations.

Proof of Lemma 4.7:

*Proof.* According to the definition (4):

$$\mathbb{E}_u \left[ \left\| \hat{\nabla}_z f_\delta^t \left( z^t \right) \right\|_2 \right]$$

$$= \mathbb{E}_u \left[ \left\| \frac{f_\delta^t \left( z^t + \mu u^t \right) - f_\delta^t \left( z^t - \mu u^t \right)}{2\mu} u^t \right\|_2 \right]$$

$$= \frac{1}{2\mu} \mathbb{E}_u \left[ \left\| \left( f_\delta^t \left( z^t + \mu u^t \right) - f_\delta^t \left( z^t - \mu u^t \right) \right) u^t \right\|_2 \right]$$

$$\overset{(1)}{\leq} \frac{1}{2\mu} \underbrace{\mathbb{E}_u \left[ \left\| \left( f^t \left( z^t + \mu u^t \right) - f^t \left( z^t - \mu u^t \right) \right) u^t \right\|_2 \right]}_{a)} + \frac{1}{2\mu} \underbrace{\mathbb{E}_u \left[ \left\| \left( \delta \left( z^t + \mu u^t \right) - \delta \left( z^t - \mu u^t \right) \right) u^t \right\|_2 \right]}_{b)},$$

$$(36)$$

where (1) use the inequality $\|a + b\|_2 \leq \|a\|_2 + \|b\|_2$ and definition (3).

Then, for a):

$$
\mathbb{E}_u \left\| \left( f^t \left( z^t + \mu u^t \right) - f^t \left( z^t - \mu u^t \right) \right) u^t \right\|_2
$$

$$
\overset{(1)}{\leq} \mathbb{E}_u \left\| \left( f^t \left( z^t + \mu u^t \right) - f^t \left( z^t \right) - \left\langle \nabla_z f^t \left( z^t \right), \mu u^t \right\rangle \right) u^t \right\|_2
$$

$$
+ \mathbb{E}_u \left\| \left( f^t \left( z^t - \mu u^t \right) - f^t \left( z^t \right) + \left\langle \nabla_z f^t \left( z^t \right), \mu u^t \right\rangle \right) u^t \right\|_2 + \mathbb{E}_u \left\| 2 \left\langle \nabla_z f^t \left( z^t \right), \mu u^t \right\rangle u^t \right\|_2
$$

$$
\overset{(2)}{\leq} 2\mathbb{E}_u \left\| \frac{L}{2} \mu^2 \left\| u^t \right\|_2^2 u^t \right\|_2 + 2\mathbb{E}_u \left\| \left\langle \nabla_z f^t \left( z^t \right), \mu u^t \right\rangle u^t \right\|_2
$$

$$
= L\mu^2 \mathbb{E}_u \left\| u^t \right\|_2^3 + 2\mu \mathbb{E}_u \left\| \left\langle \nabla_z f^t \left( z^t \right), u^t \right\rangle \right\|_2 \left\| u^t \right\|_2
$$

$$
\overset{(3)}{\leq} L\mu^2 (d+3)^{\frac{3}{2}} + 2\mu d \left\| \nabla_z f^t \left( z^t \right) \right\|_2, \tag{37}
$$

where (1) use inequality $\|a + b + c\|_2 \leq \|a\|_2 + \|b\|_2 + \|c\|_2$; (2) uses the Assumption 4.3; (3) use the Lemma 1 in Nesterov & Spokoiny (2017).

For (b), using Assumption 4.5 and the inequalities (A.1)–(A.2), we have

$$
\mathbb{E}_{u,\delta} \left[ \left\| \left( \delta \left( z^t + \mu u^t \right) - \delta \left( z^t - \mu u^t \right) \right) u^t \right\|_2 \right]
$$

$$
\leq \left( \mathbb{E}_{u,\delta} \left[ \left( \delta \left( z^t + \mu u^t \right) - \delta \left( z^t - \mu u^t \right) \right)^2 \right] \right)^{\frac{1}{2}} \left( \mathbb{E}_u \left[ \left\| u^t \right\|_2^2 \right] \right)^{\frac{1}{2}}
$$

$$
\overset{(1)}{\leq} 2\sigma d^{\frac{1}{2}},
$$

where (1) follows from (A.2), and from Lemma 1 in Nesterov & Spokoiny (2017). Finally, we take a) and b) into (19):

$$
\mathbb{E}_u \left[ \left\| \hat{\nabla}_z f_\delta^t \left( z^t \right) \right\|_2^2 \right]
$$

$$
\leq \frac{1}{2\mu} \cdot \left( L\mu^2 (d+3)^{\frac{3}{2}} + 2\mu d \left\| \nabla_z f^t \left( z^t \right) \right\|_2 \right) + \frac{1}{2\mu} \cdot 2\sigma d^{\frac{1}{2}}
$$

$$
= \frac{L\mu}{2} (d+3)^{\frac{3}{2}} + d \left\| \nabla_z f^t \left( z^t \right) \right\|_2 + \frac{\sigma d^{\frac{1}{2}}}{\mu}.
$$

$\square$

Proof of Lemma 4.8:

*Proof.*

$$
\left\| \mathbb{E}_u \left[ \hat{\nabla}_z f_\delta^t \left( z^t \right) \right] - \nabla_z f^t \left( z^t \right) \right\|_2^2
$$

$$
\leq 2 \left\| \mathbb{E}_u \left[ \hat{\nabla}_z f_\delta^t \left( z^t \right) \right] - \mathbb{E}_u \left[ \hat{\nabla}_z f^t \left( z^t \right) \right] \right\|_2^2 + 2 \left\| \mathbb{E}_u \left[ \hat{\nabla}_z f^t \left( z^t \right) \right] - \nabla_z f^t \left( z^t \right) \right\|_2^2
$$

$$
\overset{(1)}{\leq} 2 \mathbb{E}_u \left[ \left\| \frac{\delta(z^t + \mu u^t) - \delta(z^t - \mu u^t)}{2\mu} u^t \right\|_2^2 \right] + \frac{L^2 \mu^2 (d+3)^3}{2}
$$

$$
\overset{(2)}{\leq} \frac{2\sigma^2 d}{\mu^2} + \frac{L^2 \mu^2 (d+3)^3}{2}.
$$

$\square$

where (1) uses Jensen's inequality and Lemma 3 in Nesterov & Spokoiny (2017); (2) uses inequalities (A.1)–(A.2) derived from Assumption 4.5.

Proof of Lemma 4.8:

*Proof.*

$$\left\| \mathbb{E}_u \left[ \hat{\nabla}_z f_\delta^t \left( z^t \right) \right] - \nabla_z f^t \left( z^t \right) \right\|_2^2$$

$$\leq 2 \left\| \mathbb{E}_u \left[ \hat{\nabla}_z f_\delta^t \left( z^t \right) \right] - \mathbb{E}_u \left[ \hat{\nabla}_z f^t \left( z^t \right) \right] \right\|_2^2 + 2 \left\| \mathbb{E}_u \left[ \hat{\nabla}_z f^t \left( z^t \right) \right] - \nabla_z f^t \left( z^t \right) \right\|_2^2$$

$$\overset{(1)}{\leq} 2 \, \mathbb{E}_u \left[ \left\| \frac{\delta(z^t + \mu u^t) - \delta(z^t - \mu u^t)}{2\mu} u^t \right\|_2^2 \right] + \frac{L^2 \mu^2 (d+3)^3}{2}$$

$$\overset{(2)}{\leq} \frac{2\sigma^2 d}{\mu^2} + \frac{L^2 \mu^2 (d+3)^3}{2}.$$

$\square$

where (1) uses Jensen's inequality and Lemma 3 in Nesterov & Spokoiny (2017); (2) uses inequalities (A.1)–(A.2) derived from Assumption 4.5.

Combining the above bounds, we obtain the following result:

$$\sum_{t=1}^{T} \left[ \left\| \nabla_z F_{w,\alpha}^t \left( z^t \right) \right\|_2^2 \right]$$

$$\leq \frac{\left( 4H + 2V^T \right) G_\infty}{\eta} + \frac{T G_\infty}{W \epsilon^{\frac{1}{2}}} \left( \frac{2d\sigma^2}{\mu^2} + \frac{L^2 \mu^2 (d+3)^3}{2} \right)$$

$$+ \frac{L T \eta M^{\frac{1}{2}} d^{\frac{1}{2}} G_\infty}{2W(1-\gamma)\epsilon^{\frac{1}{2}}} \left( \frac{L\mu(d+3)^{\frac{3}{2}}}{2} + dG + \frac{d^{\frac{1}{2}} \sigma}{\mu} \right).$$

## B  EXPERIMENTAL SUPPLEMENTATION

To more accurately emulate real-world online scenarios, the data are streamed from the download URL to the local environment rather than preloaded. This streaming setup more faithfully reflects the conditions under which online prompt optimization methods operate. The learning rate is selected from the set $\{0.01, 0.05, 0.1, 0.2, 0.5\}$ and the zeroth-order parameter from $\{0.01, 0.05, 0.1\}$, both determined through grid search. For the INSTRUCTZERO parameters, we set the intrinsic dimension to $10$ and the prompt token length to $5$, and initialize the random projection matrix using a uniform distribution, following the original paper (Chen et al., 2023). In the adaptive uncertainty scaling mechanism, the window size is selected from the set $\{10, 30, 50, 70, 100\}$, with $\alpha$ selected from $\{0.8, 0.9, 0.95\}$ and $\beta$ from $\{0.9, 0.95, 0.99\}$.

We report all decoding and generation configurations used in our experiments. Vicuna-7B and Vicuna-13B use greedy decoding by default ($do_s ample$ = false), and although they include sampling values such as temperature = 0.6, $top - p = 0.9$, and $top - k = 50$, these values do not take effect under greedy decoding. GPT-3.5-turbo is used with its default sampling configuration (temperature = 1.0, $top - p = 1.0$). For Llama-3.1-8B, Qwen2.5-14B, and Qwen3-235B, we follow each model's default settings. For Dreamlike-Photoreal-2.0 and Stable Diffusion v1.5, which do not provide unified global defaults, we follow commonly adopted Diffusers settings, using classifier-free guidance (CFG) = 7.0, 50 sampling steps, and Euler/Euler-A samplers.

For all baseline models, we use the default parameter settings provided in their official implementations without modification.

**MANUAL PROMPT:** directly use the initial prompt without optimizing it during the process.

**ICL** (Brown et al., 2020): directly inputs the selected examples into the LLM to rewrite the original prompt, providing AOZPT with an unoptimized initial performance point.

**BDPL** (Diao et al., 2022): uses a policy gradient method to estimate the gradients of the prompt token probability distributions and employs a variance-reduced policy gradient estimator to improve training stability.

**RLPROMPT** (Deng et al., 2022): proposes a reinforcement learning-based method for optimizing discrete text prompts by training a small policy network (MLP) to generate optimized discrete prompt sequences that maximize downstream task rewards, while enhancing training stability and effectiveness through reward normalization and piecewise reward design.

**SFT** (Hao et al., 2024): performs supervised fine-tuning of a pretrained language model using 360k source-target prompt pairs (original inputs and manually optimized prompts), enabling the model to learn to generate high-quality optimized prompts from user inputs.

**Promptist** (Hao et al., 2024): builds upon SFT by further training the prompt generation policy using reinforcement learning (PPO algorithm), maximizing a reward function that combines the relevance and aesthetic scores of generated images, thereby enabling automatic exploration and generation of higher-quality prompts that better align with user intentions to improve text-to-image generation.

---

**Algorithm 2** Zeroth-order Online Gradient Descent (ZO-OGD)

---

**Input:** learning rate $\eta$, smooth parameter $\mu$, number of samples $Q$.
**Output:** $\{z^t\}_{t=1}^{T}$.
Initialize $z^0$.
**for** $t = 0$ **to** $T - 1$ **do**
    Receive $D^t = \{x^t, y^t\}$.
    Get $\{u_q^t\}_{q=1}^{Q}$ by sampled uniformly from unit sphere $\mathcal{S}^d := \{u \in \mathbb{R}^d : \|u\|_2 = 1\}$.
    Compute $f_\delta^t(z^t + \mu u_q^t)$ and $f_\delta^t(z^t - \mu u_q^t)$ by (3).
    Compute the estimation gradient $\hat{\nabla}_z f_\delta^t(z^t)$:

$$\hat{\nabla}_z f_\delta^t(z) = \frac{1}{Q} \sum_{q=1}^{Q} \frac{f_\delta^t(z^t + \mu u_q^t) - f_\delta^t(z^t - \mu u_q^t)}{2\mu} u, \tag{38}$$

    Update $z^{t+1} \leftarrow z^t - \eta \cdot \hat{\nabla}_z f_\delta^t(z^t)$.
**end for**

---

### B.1 Larger LLMs

We conduct experiments on text-to-text generation tasks using the Qwen3-235B and GPT-4o-mini model and compare AOZPT with several baseline methods. As shown in Table 4, AOZPT achieves the best performance the GSM8K dataset (accuracy), outperforming all baselines including MP, ICL, BDPL, RLPROMPT, and ZO-OGD. As shown in Table 5, AOZPT achieves the best performance on both the CNN/DailyMail dataset (F1 score) and the GSM8K dataset (accuracy), outperforming all baselines including MP, ICL, BDPL, RLPROMPT, and ZO-OGD. These results demonstrate that AOZPT consistently delivers superior performance across different text generation and reasoning tasks.

Table 4: The experiments with the Qwen3-235B model for GSM8K dataset

| Model | Method | Cumulative Accuracy |
|---|---|---|
| | MP | $83.267 \pm 0.987$ |
| | ICL | $88.133 \pm 0.833$ |
| | BDPL | $83.446 \pm 1.453$ |
| Qwen3-235B | RLPROMPT | $83.600 \pm 0.200$ |
| | ZO-OGD | $88.733 \pm 0.998$ |
| | **AOZPT** | $\mathbf{90.800 \pm 0.993}$ |

### B.2 Data drift experiments

To further emphasize this need, we have incorporated a text-to-image experiment under data-drift conditions. Specifically, we simulated a dynamic data stream by arranging samples from the Anime and Painting categories in the text-to-image task at intervals of 15, 50, 75 and 150 for Stable Diffusion

Table 5: Performance comparison of different methods on CNN/DailyMail and GSM8K datasets using GPT-4o-mini model.

| Method | CNN/DailyMail (F1 score) | GSM8K (Accuracy) |
|---|---|---|
| MP | 25.424 ± 0.171 | 86.400 ± 1.587 |
| ICL | 29.254 ± 0.187 | 91.048 ± 0.641 |
| BDPL | 26.246 ± 0.184 | 90.314 ± 0.116 |
| RLPROMPT | 27.363 ± 0.059 | 89.867 ± 1.206 |
| ZO-OGD | 30.308 ± 0.142 | 90.533 ± 1.007 |
| AOZPT | **31.016 ± 0.058** | **91.667 ± 0.757** |

v1.5 model, the results demonstrate that under varying degrees of data drift (L = 10, 50, 75, 150), the online black-box optimization algorithm, ZO-OGD and AOZPT, consistently achieves higher accuracy than traditional baselines, including MP, ICL, SFT, and Promptist.

Table 6: Data drift experiments with multiple intervals (L) for Stable Diffusion v1.5 model.

| Method | Average aesthetic quality | Method | Average aesthetic quality |
|---|---|---|---|
| MP | 5.597 ± 0.007 | ZO-OGD (L=50) | 6.134 ± 0.015 |
| ICL | 5.892 ± 0.013 | AOZPT (L=50) | **6.143 ± 0.014** |
| SFT | 5.862 ± 0.016 | ZOOGD (L=75) | 6.115 ± 0.007 |
| Promptist | 5.795 ± 0.011 | AOZPT (L=75) | **6.126 ± 0.014** |
| ZO-OGD (L=10) | 6.092 ± 0.015 | ZO-OGD (L=150) | 6.037 ± 0.080 |
| AOZPT (L=10) | **6.110 ± 0.024** | AOZPT (L=150) | **6.117 ± 0.010** |

## B.3 ABLATION STUDY

We added ablation experiments in Table 7 and Table 8: The results show that due to the high variance of zero-order optimization and the output uncertainty of generative models, the performance improvement of online zero-order prompt tuning is limited. However, after incorporating our proposed Adaptive Uncertainty Scale Adjustment mechanism, the performance improvement becomes more pronounced. $\Delta_1$ denotes the Adaptive Uncertainty Scale Adjustment mechanism and $\Delta_2$ denotes online zero-order prompt tuning.

Table 7: Ablation Study for Anime and Painting datasets. $\Delta_1$ denotes the Adaptive Uncertainty Scale Adjustment mechanism and $\Delta_2$ denotes online zero-order prompt tuning.

| Datasets | Anime | | Painting | |
|---|---|---|---|---|
| Method | Dreamlike-2.0 | Stable Diffusion v1.5 | Dreamlike-2.0 | Stable Diffusion v1.5 |
| AOZPT w/o $\Delta_1$ & $\Delta_2$ | 5.855±0.011 | 5.601±0.006 | 6.179±0.002 | 5.902±0.011 |
| AOZPT w/o $\Delta_1$ | 5.861±0.005 | 5.613±0.016 | 6.173±0.020 | 5.930±0.013 |
| AOZPT | **6.282±0.021** | **5.930±0.015** | **6.656±0.015** | **6.313±0.009** |

Table 8: Ablation Study for CNN/DailyMail and GSM8K datasets for Llama-3.1-8B model. $\Delta_1$ denotes the Adaptive Uncertainty Scale Adjustment mechanism and $\Delta_2$ denotes online zero-order prompt tuning.

| Dataset | Method | F1 score / accuracy |
|---|---|---|
| CNN/DailyMail | AOZPT w/o $\Delta_1$ & $\Delta_2$ | 23.500 ± 0.601 |
| | AOZPT w/ $\Delta_1$ | 25.089 ± 3.884 |
| | AOZPT | **27.966 ± 0.153** |
| GSM8K | AOZPT w/o $\Delta_1$ & $\Delta_2$ | 69.267 ± 0.462 |
| | AOZPT w/ $\Delta_1$ | 74.000 ± 5.415 |
| | AOZPT | **75.533 ± 0.643** |

We also conducted ablation studies on the single-point gradient estimation method in the text-to-image generation task to analyze the contribution of each component. As shown in Table 9, removing the Adaptive Uncertainty Scale Adjustment mechanism leads to a noticeable performance drop, while the full model (with consistently achieves the highest aesthetic scores on both the Anime and Painting datasets. These results indicate that the performance gains primarily come from our Adaptive Uncertainty Scale Adjustment mechanism rather than from the gradient estimation alone.

Table 9: Experimental results of the single-point method on the Anime and Painting datasets using the Stable Diffusion v1.5 model. $\Delta_1$ denotes the Adaptive Uncertainty Scale Adjustment mechanism and $\Delta_2$ denotes online zero-order prompt tuning.

| Datasets | Methods | Aesthetic |
|---|---|---|
| Anime | AOZPT (single-point method) w/o $\Delta_1$ & $\Delta_2$ | $5.710 \pm 0.021$ |
| | AOZPT (single-point method) w/o $\Delta_1$ | $5.815 \pm 0.026$ |
| | AOZPT (single-point method) | $\mathbf{5.872 \pm 0.021}$ |
| Painting | AOZPT (single-point method) w/o $\Delta_1$ & $\Delta_2$ | $6.074 \pm 0.015$ |
| | AOZPT (single-point method) w/o $\Delta_1$ | $6.184 \pm 0.028$ |
| | AOZPT (single-point method) | $\mathbf{6.219 \pm 0.016}$ |

We project optimized soft prompts onto the vocabulary via nearest-neighbor search in the embedding space. Retaining the soft-prompt configuration described in the manuscript, we replace the discrete prompts generated by the frozen open-source LLM with these projected tokens; results for LLama3.1-8B and Qwen2.5-14B models on CNN/DailyMail dataset are reported in the Table 10.

Table 10: Directly mapping experiments for LLaMA3.1-8B and Qwen2.5-14B models. "without open-source LLMs" means directly mapping the soft prompts onto the vocabulary instead of using an open-source LLM.

| Model | Method | Cumulative F1 score |
|---|---|---|
| LLaMA3.1-8B | ICL without open-source LLMs | $9.890 \pm 0.028$ |
| | ICL | $\mathbf{23.500 \pm 0.601}$ |
| | AOZPT without open-source LLMs | $9.911 \pm 0.023$ |
| | AOZPT | $\mathbf{24.707 \pm 0.047}$ |
| Qwen2.5-14B | ICL without open-source LLMs | $21.67 \pm 0.015$ |
| | ICL | $\mathbf{23.064 \pm 0.028}$ |
| | AOZPT without open-source LLMs | $21.84 \pm 0.152$ |
| | AOZPT | $\mathbf{24.767 \pm 0.502}$ |

We conduct experiments on the text-to-image generation task using different open-source LLMs, including WizardLM-13B and OpenChat-3.5-0106, to evaluate the generality of AOZPT. As shown in Table 11, AOZPT consistently achieves the best or near-best aesthetic scores across all LLM configurations, outperforming baseline methods such as MP, SFT, Promptist, ICL, and ZO-OGD. These results demonstrate that AOZPT performs robustly and effectively across various open-source LLMs, highlighting its strong adaptability and general applicability.

Table 11: Experiments on the Anime dataset are conducted using the Stable Diffusion v1.5 model, with WizardLM-13B and OpenChat-3.5-0106 serving as the open-source LLMs.

| Non-LLM | | WizardLM-13B | | openchat-3.5-0106 | |
|---|---|---|---|---|---|
| Method | Aesthetic | Method | Aesthetic | Method | Aesthetic |
| MP | $5.336 \pm 0.010$ | ICL | $5.515 \pm 0.017$ | MP | $5.479 \pm 0.003$ |
| SFT | $5.621 \pm 0.025$ | ZO-OGD | $5.635 \pm 0.053$ | ZO-OGD | $5.710 \pm 0.081$ |
| Promptist | $5.579 \pm 0.006$ | AOZPT | $\mathbf{5.734 \pm 0.028}$ | AOZPT | $\mathbf{5.828 \pm 0.064}$ |

## B.4 AOZPT vs. Adaptive Gradient Algorithm

To overcome this limitation of Adam-like algorithms with all historical gradients, we introduce a forgetting window mechanism. This approach uses an adjustable sliding window to focus on the most recent data, enabling better adaptation to dynamic input streams. Theoretically, the proposed AOZPT algorithm exhibits sublinear regret convergence. In experiments, we compare the performance of Adam, Nadam, RMSProp with AOZPT across various window sizes (w = 10, 20, 50) using the Anime and Painting dataset under a new experimental setup. The experimental results ( Table 12 ) demonstrate that by appropriately adjusting the sliding window size, the performance of AOZPT consistently outperforms the Adam, Nadam, and RMSProp algorithms. Moreover, in the majority of cases, the AOZPT algorithm with the sliding window configuration yields optimal performance.

Table 12: Performance comparison across adaptive gradient algorithms and AOZPT with varying window size.

| Datasets | Anime | | Painting | |
|---|---|---|---|---|
| Method | Dreamlike-2.0 | Stable Diffusion v1.5 | Dreamlike-2.0 | Stable Diffusion v1.5 |
| Adam Kaya et al. (2023) | $5.866 \pm 0.007$ | $5.609 \pm 0.023$ | $6.179 \pm 0.011$ | $5.927 \pm 0.025$ |
| Nadam Diederik (2014) | $5.863 \pm 0.005$ | $5.594 \pm 0.015$ | $6.168 \pm 0.009$ | $5.929 \pm 0.005$ |
| RMSProp Zou et al. (2019) | $5.860 \pm 0.007$ | $5.608 \pm 0.024$ | $6.168 \pm 0.013$ | $5.924 \pm 0.022$ |
| AOZPT ($w = 10$) | $5.879 \pm 0.016$ | $5.616 \pm 0.026$ | $6.140 \pm 0.034$ | $5.928 \pm 0.011$ |
| AOZPT ($w = 20$) | $\mathbf{5.881 \pm 0.005}$ | $5.617 \pm 0.012$ | $6.180 \pm 0.013$ | $\mathbf{5.938 \pm 0.012}$ |
| AOZPT ($w = 50$) | $5.871 \pm 0.008$ | $\mathbf{5.621 \pm 0.003}$ | $\mathbf{6.181 \pm 0.017}$ | $5.935 \pm 0.011$ |

## B.5 De–En translation task

We additionally evaluate the methods on the English-to-German (De–En) translation task using both GPT-4o-mini and Llama-3.1-8B, evaluated with the BLEU score. As shown in Table 13, AOZPT achieves the highest BLEU scores across both models, outperforming MP, ICL, and ZO-OGD. These results indicate that AOZPT delivers the best performance on the machine translation task as well.

Table 13: Experiments on the WMT/WMT14 De–En translation task using the GPT-4o-mini and Llama-3.1-8B models, evaluated with the BLEU score.

| Model | MP | ICL | ZO-OGD | AOZPT |
|---|---|---|---|---|
| GPT-4o-mini | $37.651 \pm 0.172$ | $37.929 \pm 0.349$ | $37.697 \pm 0.253$ | $\mathbf{38.975 \pm 0.195}$ |
| Llama-3.1-8B | $30.498 \pm 0.221$ | $30.728 \pm 1.110$ | $30.722 \pm 1.629$ | $\mathbf{32.510 \pm 0.334}$ |

## B.6 Parameters Sensitivity Analysis

We conducted a parameter sensitivity analysis of the AOZPT algorithm on the text-to-image generation task, examining key hyperparameters including the learning rate ($\eta$), smooth parameter ($\mu$), sliding window ($w$), weighting parameter ($\alpha$ and $\beta$). As shown in Figure 3, although the curves exhibit some local fluctuations across different parameter ranges, the overall performance consistently remains at a high level without any significant degradation. These results demonstrate that AOZPT exhibits strong robustness to hyperparameter variations, maintaining high-quality generation performance across a wide range of configurations without requiring precise parameter tuning.

## B.7 Inference latency and memory consumption

We evaluate the training and inference latency as well as memory consumption for text-to-text tasks on the Llama-3.1-8B model, as summarized in Table 14. During the training phase, AOZPT and ZO-OGD exhibit higher latency and memory usage due to the cost of performing zero-order gradient estimation. However, during inference, our AOZPT approach does not require additional computation. The semantic-rich prompts generated by the open-source LLM can be directly concatenated with the input sequence and fed into the target model, resulting in no additional inference latency or memory

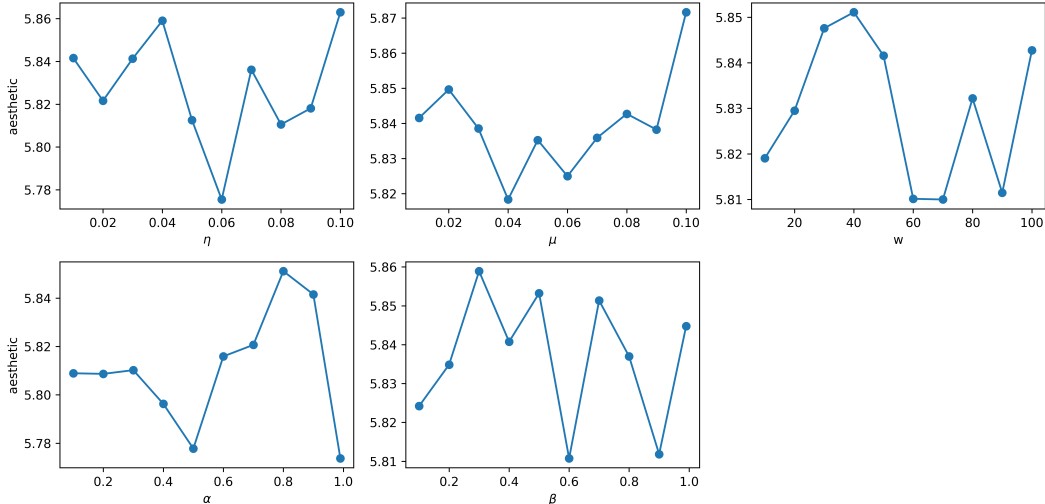

Figure 3: Parameter sensitivity experiments using Stable Diffusion v1.5 in Anime dataset: learning rate ($\eta$), smooth parameter ($\mu$), sliding window ($w$), weighting parameter ($\alpha$ and $\beta$).

consumption. In summary, although AOZPT introduces higher cost during training, it maintains efficient inference with negligible overhead, making it practical for real-world deployment.

Table 14: The inference latency and memory of the AOZPT algorithm on the Llama-3.1-8B model for text-to-text tasks.

| Datasets | Methods | Train Latency (s) | Train Memory (MiB) | Inference Latency (s) | Inference Memory (MiB) |
|---|---|---|---|---|---|
| CNN/DailyMail | MP | - | - | 6.8668 | 16592 |
| | ICL | 4.8849 | 42108 | 6.7638 | 15595 |
| | ZO-OGD | 16.8986 | 41972 | 6.8645 | 16542 |
| | AOZPT | 17.6268 | 42324 | 6.9658 | 15492 |
| GSM8K | MP | - | - | 4.9864 | 15902 |
| | ICL | 7.6808 | 41824 | 4.7865 | 15904 |
| | ZO-OGD | 24.6097 | 41828 | 4.9987 | 15933 |
| | AOZPT | 24.4819 | 41826 | 4.8687 | 15722 |

## B.8 NEW ONLINE PROMPT OPTIMIZATION BASELINE

To further contextualize AOZPT's performance, we additionally evaluated it against ACING Kharrat et al. (2025), a recently proposed reinforcement-learning–based instruction-optimization method, and adapted it to our online interaction setup for a consistent comparison. The results (Table 15) show that AOZPT achieves superior performance on text-generation tasks under this online setting, demonstrating its effectiveness in dynamic, real-time scenarios. We plan to incorporate more state-of-the-art online baselines in future work.

Table 15: The supplementary baseline experiments conducted with GPT-3.5-turbo on the CNN/DailyMail and GSM8K datasets.

| Method | CNN/DailyMail (F1 score) | GSM8K (Accuracy) |
|---|---|---|
| ACINGKharrat et al. (2025) | $28.632 \pm 2.225$ | $72.746 \pm 2.376$ |
| AOZPT (Ours) | $\mathbf{35.399 \pm 0.297}$ | $\mathbf{78.133 \pm 3.583}$ |

## B.9 CASE STUDY

Table 16: Case study of the images generated by Dreamlike-photoreal-2.0 model, where these images are generated based on the original and optimized prompts. We generate 3 images for each prompt.

| Original Prompt | Optimized Prompt |
|---|---|
| There is an image that represents the balance between yin and yang. | There is an image that represents the balance between yin and yang, harmonious, balanced, complementary, contrasting, dynamic, equilibrium, opposing forces, yin-yang symbol, balance of nature, yin-yang theory, traditional Chinese art, digital painting, artstation, concept art, smooth, sharp focus, illustration. |

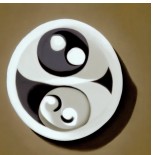 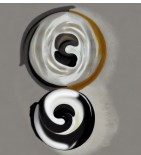 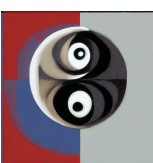 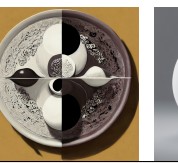 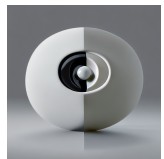 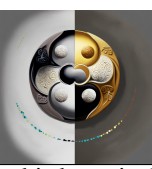

| Original Prompt | Optimized Prompt |
|---|---|
| Portrait of Herzl as a florist. | A floral portrait of Herzl, with a focus on his botanical interests and the beauty of nature. The image is highlydetailed and intricate, with a smooth and sharp focus on Herzl's face and the flowers he is holding. The background is soft and subtle, with a hint of hdri lighting to create a sense of depth and dimension. The overall style is elegant and sophisticated, with a touch of fantasy and imagination. |

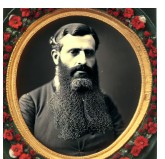 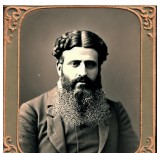 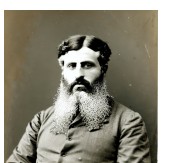 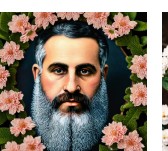 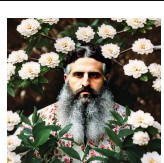 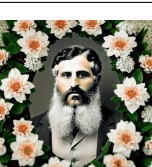

| Original Prompt | Optimized Prompt |
|---|---|
| A group of fairies playing cards on a table in a moonlit forest next to a pond filled with water lilies, artwork by Ida Rentoul Outhwaite. | A group of fairies playing cards on a table in a moonlit forest next to a pond filled with water lilies, digital painting, artstation, concept art, soft light, hdri, smooth, sharp focus, illustration, fantasy, inspired by the artwork of Ida Rentoul Outhwaite. |

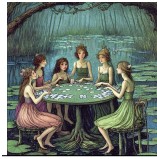 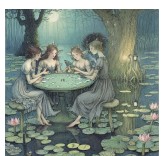 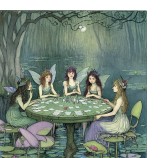 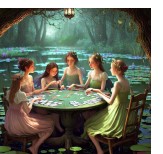 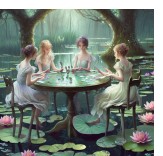 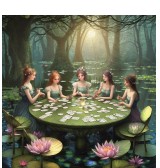

| Original Prompt | Optimized Prompt |
|---|---|
| A serene landscape depicting a garden of Eden with lake reflections, fruit trees, and animals, captured in vivid and psychedelic style. | A serene landscape depicting a garden of Eden with lake reflections, fruit trees, and animals, captured in vivid and psychedelic style, digital painting, artstation, concept art, soft light, hdri, smooth, sharp focus, illustration, fantasy. |

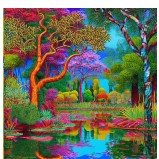 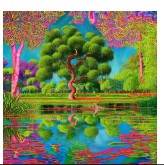 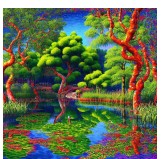 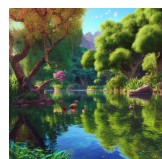 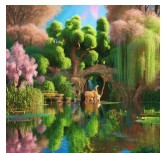 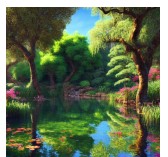

Table 17: Case study of the images generated by Stable Diffusion v1.5 model, where these images are generated based on the original and optimized prompts. We generate 3 images for each prompt.

| Original Prompt | Optimized Prompt |
| --- | --- |
| Young wizard practicing a spell while holding a spell book and a black ball in a large room, wearing intricate leather armor, in a comic cover art style with a plain background. | A young wizard is practicing a spell while holding a spell book and a black ball in a large room, wearing intricate leather armor, in a comic cover art style with a plain background, digital painting, artstation, concept art, soft light, hdri, smooth, sharp focus,illustration, fantasy. |

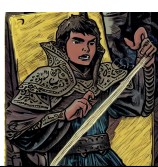 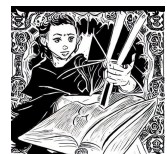 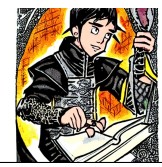 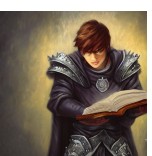 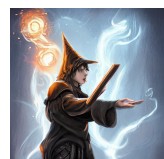 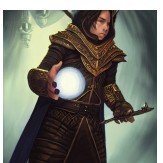

| A neoclassic painting of a box of radiation featured on ArtStation. | A neoclassic painting of a box of radiation, digital art, ArtStation, featuring a unique and intricate design, with smooth and sharp focus, creating a sense of depth and dimension. The painting is highly detailed and elegant, showcasing the artist's creativity and skill. The use of soft light and HDRi creates a sense of realism and atmosphere, transporting the viewer into the world of the painting. |

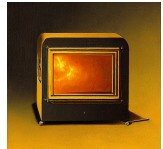 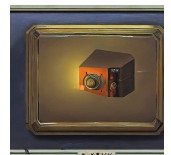 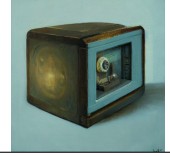 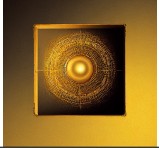 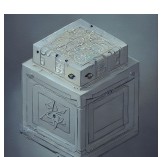 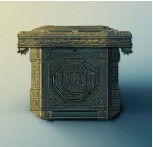

| Description, An artistic rendering of a cosmic portal with a beach at dusk on the other side. | A cosmic portal with a beach at dusk on the other side, digital painting, artstation, concept art, soft light, hdri, smooth, sharp focus, illustration, fantasy, surrealism. |

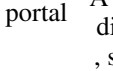

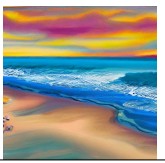 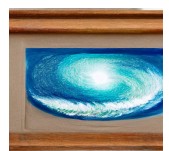 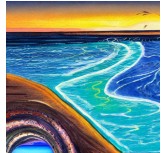 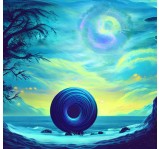 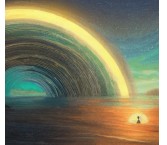 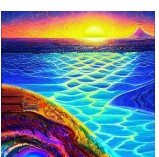

| A movie poster featuring chicken, cow, capybara, and pig in an epic cinematic style. | A movie poster featuring chicken, cow, capybara, and pig in an epic cinematic style, digital painting, artstation, concept art, highly detailed, smooth, sharp focus, illustration, fantasy, bold colors, dynamic composition, inspired by classic movie posters. |

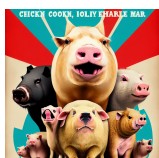 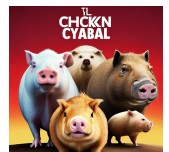 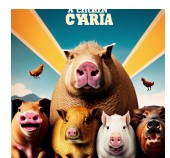 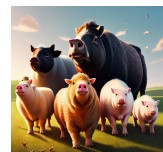 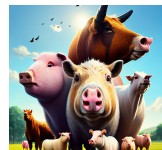 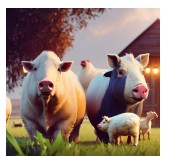

## C    ADDITIONAL EXAMPLES OF ONLINE BLACK-BOX PROMPT OPTIMIZATION

To further illustrate the practical applicability of our method, we present **additional examples from high-stakes domains such as healthcare, finance, and law, where the feature distribution of input data is rarely stationary**. Instead, it evolves continuously due to external factors.

In healthcare, for example, the emergence of new diseases, viral mutations, and updates to clinical guidelines can shift the statistical properties of diagnostic data. In finance, market volatility, policy changes, and geopolitical events may rapidly alter user behavior and transaction patterns. In the legal domain, regulatory revisions, judicial reinterpretations, and evolving precedent can significantly affect document analysis and compliance workflows. Collectively, these dynamic factors contribute to data drift—a phenomenon where previously effective prompts become misaligned with current data distributions.

Data drift poses a substantial challenge for prompt-based language models: prompts that once yielded reliable outputs may no longer meet evolving task requirements, leading to degraded performance or even high-risk errors. To maintain model reliability in such non-stationary environments, prompts must be continually adapted to reflect changes in user needs and input characteristics. This necessitates online learning capabilities during deployment.

However, many real-world applications—such as clinical decision-support systems, enterprise compliance tools, and mobile-edge devices—operate in resource-constrained settings that lack the computational capacity for backpropagation-based fine-tuning. In such environments, traditional gradient-based methods are impractical.

To address this limitation, we propose online black-box prompt optimization as a lightweight yet effective alternative. This approach does not require access to model gradients or internals. Instead, it leverages expert feedback to iteratively refine prompts. For example, physicians can assess the accuracy of generated diagnoses, auditors may flag anomalous transactions, and legal professionals can evaluate or correct machine-generated legal advice. These expert feedback signals serve as a supervisory signal, enabling models to adapt prompts in real time—without backpropagation—to maintain robustness in the presence of streaming, non-stationary data.

## D    USE OF LLMS

In this work, LLMs are employed solely for polishing or grammar checking text that is originally written by us.

