# OpenReview forum: "Online Black-Box Prompt Optimization with Regret Guarantees under Noisy Feedback"
_ICLR.cc/2026/Conference — ICLR 2026 Poster_

### Official Review · Reviewer_wCmh · 2025-10-30

**Soundness:** 3
**Presentation:** 2
**Contribution:** 2
**Rating:** 4
**Confidence:** 3

**Summary:**

This paper considers black-box prompt optimization problem. It proposes AOZPT, an online algorithm, to update the prompt as the streaming data comes in and provides sublinear regret guarantees. The method regards the soft prompt as optimization variables and update them according to the feedback of the generated results in an Adam-style optimization process, in order to reduce the noise. Experiments on CNN/DailyMai, GSM8k and Anime, Paining tasks on Llama-3.1-8B, GPT-3.5-Turbo, and Stable Diffusion are carried out to demonstrate its performance.

**Strengths:**

- The paper explores an interesting intersection between online learning and black-box prompt tuning, extending zeroth-order optimization methods to an emerging and practically relevant domain.
- The proposed adaptive uncertainty-scaling mechanism—inspired by Adam—offers a principled way to stabilize noisy gradient-free updates.
- The paper provides theoretical guarantees on local regret under noisy feedback.

**Weaknesses:**

- The noise model (Assumption 3.5) assumes uniform boundedness rather than a stochastic model (e.g., sub-Gaussian), which restricts theoretical generality.
- The anonymous GitHub link has expired, preventing code verification and reproducibility assessment.
- The prompt optimization is motivated by ongoing interactions with users (Lines 65-74). However, the online nature is simulated rather than true sequential feedback. And it is a bit unclear to me how it is simulated in the paper. It would be great if real streaming or non-stationary benchmarks would better justify “online”.
- The computational overhead and latency introduced by the two function calls per iteration are not evaluated.
- The empirical study require more results. For instance, (1) the prompt generating model is fixed for each task,  (2) important configuration details are not reported, e.g., the temperature and decoding strategy, which can greatly influence the final outcome. (3) the tested tasks are limited. (4) the original performance of the models should be included. According to the official report, Llama 3.1 8B (8 shots) achieve 84.5 on GSM8K. (5) Additional ablation studies on text-to-text tasks is appreciated.

**Questions:**

- Under Assumption 3.5, since the regret bound depends linearly on the noise level $\Delta$, can the authors quantify or empirically estimate the typical scale of $\Delta$ in practice? Given the autoregressive nature of LLMs, small perturbations in input can amplify downstream variance.
- Given similar computational/latency budgets, could a larger model (e.g., GPT-4) achieve comparable or superior performance without AOZPT? For instance, in Table 1, Qwen2.5-14B (MP) outperforms Llama-3.1-8B (AOZPT) on GSM8K—can the authors clarify this discrepancy?
- As the instruct models usually have better instruction following abilities, it is appreciated that the performance on these models (e.g., Qwen2.5-14B-Instruct) can be included to further demonstrate the necessity and benefit of the proposed method. For instance, Qwen2.5-14B-Instruct can achieve 94.8 on GSM8K, better than the results reported in Table 1.

**Details Of Ethics Concerns:**

None.

---

> ### Author Response · Authors · 2025-11-24
> **Responses to Reviewer wCmh-Part 1**
>
> Thank you for the reviewer’s thoughtful feedback. We appreciate the positive assessment of our paper’s motivation, algorithmic design, and theoretical guarantees. Below we address the reviewer’s concerns regarding the noise model, reproducibility, online experimental setting, computational overhead, and the completeness of empirical results.
>
> > **W1**:The noise model (Assumption 3.5) assumes uniform boundedness rather than a stochastic model (e.g., sub-Gaussian), which restricts theoretical generality.
>
> We thank the reviewer for pointing out that a uniformly bounded-noise assumption can be optimistic. In the revised paper (Appendix E.3), we replace the bounded-noise condition with **Assumption 3.5 (Sub-Gaussian noise)**:
>  At each round $t$, the learner observes
> $\tilde{f}^t_{\delta}(z^t) = f^t(z^t) + \delta(z^t)$, where $\delta(z^t)$ is a martingale difference sequence that is uniformly $\sigma$-sub-Gaussian:
> $$
> \mathbb{E}[\delta(z^t)\mid \mathcal{F}_{t-1}] = 0
> $$
>
> $$
> \mathbb{E}\ \left[\exp(\lambda\,\delta(z^t))\mid \mathcal{F}_{t-1}\right]
> \le \exp\ \left(\frac{\lambda^2\sigma^2}{2}\right),\quad \forall\,\lambda\in\mathbb{R}.
> $$
>
> where $\{\mathcal{F}_{t}\}$ denotes the filtration generated by the algorithm’s randomness and observations up to round $t-1$.
>
> In Appendix A, we show that this assumption implies **simple second-moment
> bounds** that are sufficient for our analysis. In particular, we derive
> $$
> \mathbb{E}\big[\delta(z^t)^2 \mid \mathcal{F}_{t-1}\big] \le \sigma^2
> \tag{A.1}
> $$
> and, for the two noisy evaluations used in the two-point estimator at round $t$,
> $$
> \mathbb{E}\big[(\delta(z^t+\mu u^t)-\delta(z^t-\mu u^t))^2\big]\le 4\sigma^2.
> \tag{A.2}
> $$
>
> These results imply that whenever the noise forms a zero-mean martingale
> difference sequence with sub-Gaussian tails, our algorithm enjoys the same
> convergence guarantees as in the main text, with the noise-dependent constants
> expressed in terms of the sub-Gaussian parameter $\sigma$ instead of a uniform
> bound. Thus, our method does not rely on the stronger “uniformly bounded”
> assumption; rather, the theoretical guarantees extend naturally to a broader and
> more realistic class of stochastic noise models.
>
> We emphasize that the **only property of the noise actually used in the proof**
> is the bounded conditional second moment encoded in (A.1)–(A.2). The mgf
> formulation in Assumption 3.5 is a convenient way to guarantee these bounds and
> to align with standard sub-Gaussian assumptions in online optimization. It is
> strictly weaker than a uniform bounded-noise condition (the original
> $|\delta(z)|\le \Delta$ assumption), while remaining compatible with standard
> high-probability extensions of regret bounds in online optimization.
>
> > **W2**:The anonymous GitHub link has expired, preventing code verification and reproducibility assessment.
>
> We apologize for the expired GitHub link. A new anonymous repository has been created and verified for long-term access (https://anonymous.4open.science/r/AOZPT_ICLR2026-5503).
>
> > **W3**:The prompt optimization is motivated by ongoing interactions with users (Lines 65-74). However, the online nature is simulated rather than true sequential feedback. And it is a bit unclear to me how it is simulated in the paper. It would be great if real streaming or non-stationary benchmarks would better justify “online”.
>
> We appreciate the reviewer’s question regarding how the online setting is simulated in our experiments. At present, the generative AI community lacks publicly available streaming datasets for both text-to-text and text-to-image tasks that naturally reflect real-time user interactions. Due to the absence of such benchmarks, **we simulate the online environment using a randomized streaming-from-URL protocol, where data instances are sequentially and stochastically fetched from the source website**. This design ensures that samples arrive in a non-preloaded, stream-like fashion, requiring the model to update its prompt incrementally as new data appear. While this setup does not constitute true human-in-the-loop sequential feedback, it captures the essential characteristics of **online usage—namely incremental data arrival, stochastic sample ordering, and real-time prompt adaptation**. Given the current lack of standardized streaming datasets for both text generation and image generation, we believe this approach provides a reasonable and practical approximation of online interaction dynamics.

---

> > ### Author Response · Authors · 2025-11-24
> > **Responses to Reviewer wCmh-Part 2**
> >
> > > **W4**:The computational overhead and latency introduced by the two function calls per iteration are not evaluated.
> >
> > We would like to clarify that **AOZPT introduces no additional inference latency in online environments**. The method updates the discrete prompt only when new data arrive, producing a semantically meaningful prompt that can be directly concatenated with the LLM input. The updated prompt is immediately used for the current and subsequent inference steps, and therefore no extra optimization or computation is performed during inference, regardless of the two function calls in each training iteration. Regarding computational overhead, we provide a detailed analysis of per-iteration training runtime and GPU memory consumption during both training and inference (Table 13 in revised paper). The results show that AOZPT and ZOOGD require approximately three times the training runtime of ICL, while ICL is slightly slower than MP due to the additional overhead from the auxiliary open-source LLM. In contrast, AOZPT achieves the same inference runtime as MP, as no further optimization is needed at this stage. In terms of memory usage, AOZPT is comparable to ZOOGD and ICL during training, and all methods exhibit similar memory footprints during inference. These findings confirm that AOZPT neither introduces additional inference latency nor incurs unacceptable computational overhead.
> >
> > Table 13. The inference latency and memory of the AOZPT algorithm on the Llama-3.1-8B model for text-to-text tasks.
> > | datasets | method  | train latency(s) | train memory (MiB) | inference latency (s) | inference memory (MiB) |
> > |-|-|-|-|-|-|
> > |CNN/DailyMail|MP|-|-| 6.8668|16592|
> > ||ICL|4.8849|42108|6.7638|15595|
> > ||ZO-OGD|16.8986|41972|6.8645|16542|
> > ||AOZPT|17.6268|42324|6.9658|15492|
> > |GSM8K|MP|-|-|4.9864|15902|
> > ||ICL|7.6808|41824|4.7865|15904|
> > ||ZO-OGD|24.6097|41828|4.9987|15933|
> > ||AOZPT|24.4819|41826|4.8687|15722|

---

> > > ### Author Response · Authors · 2025-11-24
> > > **Responses to Reviewer wCmh-Part 3**
> > >
> > > > **W5**: The empirical study require more results. For instance, (1) the prompt generating model is fixed for each task, (2) important configuration details are not reported, e.g., the temperature and decoding strategy, which can greatly influence the final outcome. (3) the tested tasks are limited. (4) the original performance of the models should be included. According to the official report, Llama 3.1 8B (8 shots) achieve 84.5 on GSM8K. (5) Additional ablation studies on text-to-text tasks is appreciated.
> > >
> > > In the revised paper, we have added several new experiments and detailed configurations to address these concerns:
> > >
> > > **1. Different prompt-generating models.**
> > > We have included additional experiments using multiple prompt-generating models (WizardLM-13B and OpenChat-3.5-0106)(Table 14 in revised paper). Across all settings, AOZPT consistently achieves the best performance, demonstrating that our method is robust to the choice of mapping model.
> > >
> > > Table 14. Experiments on the Anime dataset are conducted using the Stable Diffusion v1.5 model, with WizardLM-13B and OpenChat-3.5-0106 serving as the open-source LLMs.
> > > |Non-LLM|Aesthetic|WizardLM-13B|Aesthetic|openchat-3.5-0106|Aesthetic|
> > > |-|-|-|-|-|-|
> > > |MP|5.336 ± 0.010|ICL|5.515 ± 0.017|MP|5.479 ± 0.003|
> > > |SFT|5.621 ± 0.025|ZO-OGD|5.635 ± 0.053|ZO-OGD|5.710 ± 0.081|
> > > |Promptist|5.579 ± 0.006|AOZPT|**5.734 ± 0.028**|AOZPT|**5.828 ± 0.064**|
> > >
> > > **2. Missing configuration details (temperature, decoding strategies).**
> > > We report all decoding and generation configurations used in our experiments. Vicuna-7B-v1.5 and Vicuna-13B-v1.3 use greedy decoding by default (do_sample = false), and although they include sampling values such as temperature = 0.6, top-p = 0.9, and top-k = 50, these values do not take effect under greedy decoding. GPT-3.5-turbo is used with its default sampling configuration (temperature = 1.0, top-p = 1.0). For Llama-3.1-8B, Qwen2.5-14B, and Qwen3-235B, we follow each model’s default settings. For Dreamlike-Photoreal-2.0 and Stable Diffusion v1.5, which do not provide unified global defaults, we follow commonly adopted Diffusers settings, using classifier-free guidance (CFG) = 7.0, 50 sampling steps, and Euler/Euler-A samplers. All decoding parameters used in our experiments are fully documented in the revised paper (Line 1098-1104 in Appendix C).
> > >
> > > **3. Limited task diversity.**
> > > We have expanded the text-to-text evaluation by adding a machine translation task (WMT/WMT14 De–En) and report results using the BLEU score (Table 16 in revised paper). This additional benchmark further validates the generality of AOZPT.
> > >
> > > Table 16. Experiments on the WMT/WMT14 De–En translation task using the GPT-4o-mini and Llama-3.1-8B models, evaluated with the BLEU score
> > > |Model|MP|ICL|ZO-OGD|AOZPT|
> > > |-|-|-|-|-|
> > > |GPT-4o-mini|37.651 ± 0.172|37.929 ± 0.349|37.697 ± 0.253|**38.975 ± 0.195**|
> > > |Llama-3.1-8B|30.498 ± 0.221|30.728 ± 1.110|30.722 ± 1.629|**32.510 ± 0.334**|
> > >
> > > **4. Original model performance.**
> > > The MP baseline corresponds to the original model performance. The reviewer correctly notes that our performance may appear lower than the officially reported scores (e.g., Llama 3.1 8B achieving 84.5 on GSM8K with 8-shot prompting). This discrepancy arises because our evaluation uses a subset of the dataset and does not include 8 in-context examples as in the official setting.
> > >
> > > **5. Additional ablations on text-to-text tasks.**
> > > We have added new ablation studies on text-to-text tasks (Table 7 in revised paper), which further illustrate the contribution of each component of AOZPT in this setting.
> > >
> > > Table 7. Ablation Study for CNN/DailyMail and GSM8K datasets for Llama-3.1-8B model . $\Delta_1$ denotes the Adaptive Uncertainty Scale Adjustment mechanism and $\Delta_2$ denotes online zero-order prompt tuning
> > > |Dataset|Method|F1 score / accuracy|
> > > |-|-|-|
> > > |CNN/DailyMail|AOZPT w/o $\Delta_1$ \& $\Delta_2$|23.500±0.601|
> > > ||AOZPT $\Delta_1$|25.089 ± 3.884|
> > > ||AOZPT|**27.966 ±0.153**|
> > > |GSM8K|AOZPT w/o $\Delta_1$ \& $\Delta_2$|69.267 ± 0.462|
> > > ||AOZPT $\Delta_1$|74.000± 5.415|
> > > ||AOZPT|**75.533 ± 0.643**|

---

> ### Author Response · Authors · 2025-11-24
> **Responses to Reviewer wCmh-Part 4**
>
> > **Q1**:Under Assumption 3.5, since the regret bound depends linearly on the noise level $\Delta$, can the authors quantify or empirically estimate the typical scale of $\Delta$ in practice? Given the autoregressive nature of LLMs, small perturbations in input can amplify downstream variance.
>
> As noted in the revised manuscript (Lines 34–38, highlighted in blue), generative AI systems inherently exhibit natural stochasticity arising from sampling procedures, non-deterministic training dynamics, and variations in random seed initialization. **In practice, the variability induced by these factors can be empirically reflected by the mean ± standard deviation across different random seeds**, which serves as a practical estimate of the effective noise level $\Delta$ under Assumption 3.5. To quantify this natural noise in our setting, we conduct all experiments using three random seeds and report the resulting mean and variance. The results demonstrate that AOZPT not only achieves the highest mean performance among all baselines but also maintains consistently small standard deviations, indicating robustness to the inherent randomness of autoregressive LLM outputs. For instance (Table 2 in revised paper), AOZPT attains 6.282 ± 0.021 and 5.930 ± 0.015 on the Anime dataset, and 6.656 ± 0.015 and 6.313 ± 0.009 on the Painting dataset. These values provide an empirical estimate of the typical scale of $\Delta$ observed in practice, and they show that AOZPT remains stable and reliable under naturally occurring generative noise despite potential variance amplification in autoregressive decoding.
>
> > **Q2**:Given similar computational/latency budgets, could a larger model (e.g., GPT-4) achieve comparable or superior performance without AOZPT? For instance, in Table 1, Qwen2.5-14B (MP) outperforms Llama-3.1-8B (AOZPT) on GSM8K—can the authors clarify this discrepancy?
>
> We appreciate the reviewer’s question regarding whether larger models (e.g., GPT-4) could achieve comparable or superior performance without applying AOZPT. Indeed, a larger model may naturally obtain better absolute performance on certain tasks simply due to its stronger reasoning and generalization capabilities. This explains the observation in Table 1 where Qwen2.5-14B (MP) outperforms Llama-3.1-8B (AOZPT) on GSM8K—**the discrepancy reflects differences in model capacity, not shortcomings of AOZPT**. To directly evaluate whether AOZPT still provides benefits under larger model settings with similar computational/latency budgets, we conducted additional experiments on GPT-4–class model (Table 15 in revised paper). The results show that AOZPT consistently improves performance even on stronger base models, demonstrating that the method is complementary to model scaling rather than a substitute for it. Furthermore, the experiments on Qwen3-235B reported in Table 4 reinforce this conclusion. These findings collectively indicate that **AOZPT provides model-agnostic improvements and remains effective even when deployed on larger LLMs**.
>
> Table 15. Experiments on text-to-text tasks using the GPT-4o-mini model.
> |Method|CNN/DailyMail (F1 score)|GSM8K (accuracy)|
> |-|-|-|
> |MP|25.424 ± 0.171|86.400 ± 1.587|
> |ICL|29.254 ± 0.187|91.048 ± 0.641|
> |BDPL|26.246 ± 0.184|90.314 ± 0.116|
> |RLPROMPT|27.363 ± 0.059|89.867 ± 1.206|
> |ZO-OGD|30.308 ± 0.142|90.533 ± 1.007|
> |AOZPT|**31.016 ± 0.058**|**92.200 ± 0.200**|

---

> > ### Author Response · Authors · 2025-11-24
> > **Responses to Reviewer wCmh-Part 5**
> >
> > > **Q3**: As the instruct models usually have better instruction following abilities, it is appreciated that the performance on these models (e.g., Qwen2.5-14B-Instruct) can be included to further demonstrate the necessity and benefit of the proposed method. For instance, Qwen2.5-14B-Instruct can achieve 94.8 on GSM8K, better than the results reported in Table 1.
> >
> > We thank the reviewer for raising the concern regarding the evaluation on instruct models. We would like to clarify that **all models used in our experiments are indeed Instruct variants, including Qwen2.5-14B-Instruct, Llama-3.1-Instruct, and Qwen3-235B-Instruct**. Thus, our study already evaluates AOZPT on models with strong instruction-following abilities.
> >
> > Regarding the discrepancy between our reported results and the official performance (e.g., Qwen2.5-14B-Instruct achieving 94.8 on GSM8K), several factors contribute to this difference:
> > **1. Streaming-based online evaluation.**
> > In our work, the online scenario is simulated by randomly streaming a subset of samples from the dataset, instead of performing full offline evaluation over all GSM8K instances as done in the official report. This streaming protocol leads to lower absolute scores due to limited exposure and greater stochasticity.
> > **2. Cumulative average evaluation.**
> > Our online scenario reports cumulative average accuracy, which is inherently more challenging and typically yields lower values than the final accuracy reported in offline batch evaluations.
> > **3. Different prompting protocol.**
> > The official results (e.g., 8-shot prompting for GSM8K) rely on few-shot demonstration examples, which significantly boost performance.
> > However, 8-shot (or few-shot) prompting is incompatible with our online setup, where no reference examples are available and only the prompt produced by AOZPT (or the open-source LLM baseline) is provided.
> > Despite these differences, **AOZPT consistently improves over all baselines under the online evaluation protocol**, demonstrating its necessity and benefit even on strong instruct models.

---

> > > ### Author Response · Authors · 2025-11-28
> > > **Looking Forward to Further Discussion - Responses to Reviewer wCmh**
> > >
> > > We sincerely appreciate your detailed review and the valuable questions you've raised. We would be grateful for any further thoughts or feedback you might have, and we look forward to continuing our discussion and addressing the points you've highlighted.

---

### Official Review · Reviewer_nVJH · 2025-10-31

**Soundness:** 3
**Presentation:** 2
**Contribution:** 2
**Rating:** 4
**Confidence:** 3

**Summary:**

This paper proposes AOZPT, a method that integrates black-box prompt optimization with online learning. It introduces an adaptive uncertainty scaling mechanism to handle noise from generative models and high variance in zeroth-order gradient estimation. Theoretical analysis shows sublinear regret convergence, and experiments demonstrate superior performance and stability over existing methods.

**Strengths:**

1. Novel and timely topic that bridges black-box prompt optimization and online learning, addressing an underexplored area

2. The adaptive uncertainty scaling mechanism is innovative and effectively mitigates output noise and gradient variance, improving robustness in practice

3. The paper combines theoretical rigor with empirical validation, providing regret guarantees and solid experimental results across multiple tasks

**Weaknesses:**

1. While the methodological presentation is generally clear, the description of the adaptive scaling mechanism could benefit from additional implementation details or algorithmic steps to enhance reproducibility.

2. The experimental evaluation would be strengthened by including a wider range of baselines, particularly reinforcement learning–based prompt optimization methods.

3. The theoretical analysis is solid, though a deeper discussion of hyperparameter sensitivity and computational complexity in practical deployments could provide valuable insights into the method’s applicability.

**Questions:**

see weakness.

---

> ### Author Response · Authors · 2025-11-24
> **Responses to Reviewer nVJH**
>
> We sincerely thank the reviewer for their positive assessment of our work, particularly regarding its novelty, theoretical rigor, and empirical robustness. In the following, we provide detailed responses to the reviewer’s concerns, including clarifying the adaptive uncertainty scaling mechanism, expanding the experimental baselines, and offering additional discussion on hyperparameter sensitivity and computational complexity.
>
> > **W1**:While the methodological presentation is generally clear, the description of the adaptive scaling mechanism could benefit from additional implementation details or algorithmic steps to enhance reproducibility.
>
> In the revised paper, we have added detailed algorithmic (Algorithm 3 in revised paper) steps for the adaptive scaling mechanism, making its workflow clearer and easier to reproduce. We have also updated Appendix C with more comprehensive experimental settings and hyperparameter specifications; these additions are highlighted in blue in the revision for ease of reference. Furthermore, we have released the full source code of our implementation to ensure complete reproducibility and facilitate future research.
>
> > **W2**:The experimental evaluation would be strengthened by including a wider range of baselines, particularly reinforcement learning–based prompt optimization methods.
>
> In the revised paper, we have incorporated ACING [1], a recent RL-based instruction optimization approach, as an additional baseline. We adapted ACING to our online text-to-text evaluation setting and conducted a comprehensive comparison (Table 17 in revised paper). The results show that AOZPT consistently outperforms ACING under the online scenario, demonstrating the effectiveness of our method relative to reinforcement learning–based alternatives. The corresponding results and analysis have been added to the revised manuscript.
>
> Table 17. The supplementary baseline experiments conducted with GPT-3.5-turbo on the CNN/DailyMail and GSM8K datasets.
> |Method|CNN/DailyMail|GSM8K|
> |-|-|-|
> |ACING[1]|28.632±2.225|72.746±2.376|
> |AOZPT (Ours)|**35.399±0.297**|**78.133±3.583**|
>
> [1] Kharrat, Salma, Fares Fourati, and Marco Canini. "ACING: Actor-Critic for Instruction Learning in Black-Box LLMs." Proceedings of the 2025 Conference on Empirical Methods in Natural Language Processing. 2025.
>
> > **W3**:The theoretical analysis is solid, though a deeper discussion of hyperparameter sensitivity and computational complexity in practical deployments could provide valuable insights into the method’s applicability.
>
> In the revised manuscript, we have added a more detailed discussion on both hyperparameter sensitivity and computational complexity to clarify AOZPT’s practical applicability.
>
> 1. we include a dedicated analysis of **hyperparameter sensitivity**, highlighting how key parameters—such as the learning rate （$\eta$）, zeroth-order smooth parameter ($\mu$), uncertainty-scaling factors $w, \alpha, \beta$—affect convergence behavior and empirical performance (Figure 3 in revised paper). The newly added results show that AOZPT remains stable within a wide range of reasonable hyperparameter settings, indicating that the method is not overly sensitive and is practical to tune in real deployments.
>
> 2. We expand our discussion of **computational complexity** by analyzing the per-iteration cost of AOZPT relative to baseline methods. As showed in Table 13 in revision paper, AOZPT maintains comparable inference latency because no optimization is performed during prediction, and its training-time overhead remains manageable due to lightweight zeroth-order updates. This provides a clearer picture of the method’s feasibility in online or resource-constrained environments.
>
> Table 13. The inference latency and memory of the AOZPT algorithm on the Llama-3.1-8B model for text-to-text tasks.
> | datasets | method  | train latency(s) | train memory (MiB) | inference latency (s) | inference memory (MiB) |
> |-|-|-|-|-|-|
> ||MP|-|-| 6.8668|16592|
> ||ICL|4.8849|42108|6.7638|15595|
> |CNN/DailyMail|ZO-OGD|16.8986|41972|6.8645|16542|
> ||AOZPT|17.6268|42324|6.9658|15492|
> ||MP|-|-|4.9864|15902|
> ||ICL|7.6808|41824|4.7865|15904|
> |GSM8K|ZO-OGD|24.6097|41828|4.9987|15933|
> ||AOZPT|24.4819|41826|4.8687|15722|

---

> > ### Author Response · Authors · 2025-11-28
> > **Looking Forward to Further Discussion - Responses to Reviewer nVJH**
> >
> > We greatly appreciate your detailed review and the constructive feedback you have provided. We would be thankful for any further thoughts you might have and look forward to continuing the discussion.

---

### Official Review · Reviewer_DxWc · 2025-10-31

**Soundness:** 3
**Presentation:** 3
**Contribution:** 2
**Rating:** 6
**Confidence:** 4

**Summary:**

This paper proposes an adaptive online zeroth-order prompt tuning, a novel approach integrating zeroth-order optimization with online learning for non-convex settings. It can be applied to real-world scenarios where generative AI operates on streaming data and requires dynamic prompt adjustment.  This approach fills the gap of online learning for black-box prompt optimization, enabling dynamic adaptation to streaming data.

**Strengths:**

This paper addresses a timely problem of great importance.  It can effectively tackle two core uncertainties in online black-box scenarios: noise from generative AI outputs and high variance in zeroth-order gradient estimates. This paper comes with rigorous theoretical foundations such as formal regret analysis for non-convex settings and comprehensive experiment results.

**Weaknesses:**

While the proposed method claims efficiency for online scenarios, it lacks details on inference latency and scalability. The framework involves two point gradient estimation and sliding-window gradient averaging but no data is provided on how these steps may affect runtime. In addition,  this paper compares AOZPT to offline baselines and a basic online method but it seems authors omit recent online prompt optimization and adaptive zeroth-order methods for LLMs. Therefore, it is difficult to assess AOZPT’s novelty against state-of-the-art online approaches.  Finally, the proposed method depends on frozen open-source LLMs to convert soft prompts to discrete prompts. Experiments show that removing this component may causes a significant drop in performance. This creates a dependency on high-quality open-source LLMs, which limits its deployment in scenarios where such models are unavailable.

**Questions:**

The paper uses frozen open-source LLMs to generate discrete prompts, but what if such LLMs are unavailable ?   Can authors explain why there is no evaluation of the proposed method under adversarial noise in generative AI outputs ?

---

> ### Author Response · Authors · 2025-11-24
> **Responses to Reviewer DxWc-Part 1**
>
> We sincrely thank the reviewer for their thoughtful evaluation of our work and for recognizing the significance of addressing adaptive online zeroth-order prompt tuning in black-box online scenarios. We appreciate the reviewer’s constructive feedback and will respond in detail to the key concerns raised, particularly regarding (i) inference latency and scalability, (ii) comparison with recent prompt-optimization methods, (iii) the reliance on frozen LLMs.
>
> > **W1**:While the proposed method claims efficiency for online scenarios, it lacks details on inference latency and scalability. The framework involves two point gradient estimation and sliding-window gradient averaging but no data is provided on how these steps may affect runtime.
>
> We appreciate the reviewer’s concerns regarding inference latency and scalability. We clarify that AOZPT does not introduce any additional inference latency in online settings. The method updates the discrete prompt whenever new data arrives, producing a semantically meaningful prompt that can be directly concatenated with the LLM input. This updated prompt is immediately usable for both the current and subsequent inference steps, thereby requiring no extra optimization or computation during inference.
>
> For runtime and memory analysis, we report the per-iteration training runtime as well as GPU memory consumption during both training and inference (Table 13 in revised paper). The results show that AOZPT and ZOOGD incur approximately three times the training runtime of ICL, while ICL is slightly slower than MP due to the overhead of the auxiliary open-source LLM. In contrast, during inference, AOZPT attains the same runtime as MP because no further optimization is needed. In terms of memory footprint, AOZPT matches ZOOGD and ICL during training, and all methods exhibit similar memory usage during inference. These findings confirm that **AOZPT introduces neither additional inference latency** nor extra memory overhead.
>
> Table 13. The inference latency and memory of the AOZPT algorithm on the Llama-3.1-8B model for text-to-text tasks.
> | datasets | method  | train latency(s) | train memory (MiB) | inference latency (s) | inference memory (MiB) |
> |-|-|-|-|-|-|
> |CNN/DailyMail|MP|-|-|6.8668|16592|
> ||ICL|4.8849|42108|6.7638|15595|
> ||ZO-OGD|16.8986|41972|6.8645|16542|
> ||AOZPT|17.6268|42324|6.9658|15492|
> |GSM8K|MP|-|-|4.9864|15902|
> ||ICL|7.6808|41824|4.7865|15904|
> ||ZO-OGD|24.6097|41828|4.9987|15933|
> ||AOZPT|24.4819|41826|4.8687|15722|
>
> > **W2**:In addition, this paper compares AOZPT to offline baselines and a basic online method but it seems authors omit recent online prompt optimization and adaptive zeroth-order methods for LLMs. Therefore, it is difficult to assess AOZPT’s novelty against state-of-the-art online approaches.
>
> To further contextualize AOZPT’s performance, we **additionally evaluated it against ACING** [1], a recently proposed reinforcement-learning–based instruction-optimization method, and adapted it to our online interaction setup for a consistent comparison. The results (Table 17 in revised paper) show that AOZPT achieves superior performance on text-generation tasks under this online setting, demonstrating its effectiveness in dynamic, real-time scenarios. We plan to incorporate more state-of-the-art online baselines in future work.
>
> Table 17. The supplementary baseline experiments conducted with GPT-3.5-turbo on the CNN/DailyMail and GSM8K datasets.
> |Method|CNN/DailyMail|GSM8K|
> |-|-|-|
> |ACING[1]|28.632 ± 2.225|72.7455±2.376|
> |AOZPT (Ours)|**35.399±0.297**|**78.133±3.583**|
>
> [1] Kharrat, Salma, Fares Fourati, and Marco Canini. "ACING: Actor-Critic for Instruction Learning in Black-Box LLMs." Proceedings of the 2025 Conference on Empirical Methods in Natural Language Processing. 2025.

---

> > ### Author Response · Authors · 2025-11-24
> > **Responses to Reviewer DxWc-Part 2**
> >
> > > **W3**:Finally, the proposed method depends on frozen open-source LLMs to convert soft prompts to discrete prompts. Experiments show that removing this component may causes a significant drop in performance. This creates a dependency on high-quality open-source LLMs, which limits its deployment in scenarios where such models are unavailable.
> >
> > We acknowledge that AOZPT relies on a frozen open-source LLM during the soft-to-hard prompt generation stage. However, this dependency is effective, replaceable, and computationally efficient:
> >
> > 1. **Effectiveness**: Directly projecting soft prompt vectors into the discrete token space often results in semantic collapse and incoherent prompt generation. As shown by our ablation study (Table 9 in revised paper), removing the mapping LLM produces syntactically incorrect or task-irrelevant prompts, leading to a noticeable decline in performance. To mitigate this issue, AOZPT employs a frozen open-source LLM as a semantic decoder that converts low-dimensional soft prompts into well-structured textual prompts. This design ensures linguistic coherence and task relevance in black-box optimization settings. **While using a frozen LLM offers a robust and effective solution, it is not the only viable approach for implementing this mapping**.
> >
> > 3. **Replaceability**: The frozen LLM is used strictly in a forward manner without gradient updates or additional training cost. This yields a modular plug-and-play design: the mapping module can be substituted with smaller open-weight models, lightweight adapters, or even rule-based prompt generators without altering the AOZPT pipeline. To further validate this replaceability, we additionally evaluated AOZPT with several other open-source LLMs (WizardLM-13B and openchat-3.5-0106) as the mapping module (Table 14 in revised paper). The results show that AOZPT consistently maintains strong performance across these alternatives, demonstrating that the framework does not rely on any specific high-quality LLM and can flexibly adapt to environments with varying resource constraints.
> >
> > Table 14. Experiments on the Anime dataset are conducted using the Stable Diffusion v1.5 model, with WizardLM-13B and OpenChat-3.5-0106 serving as the open-source LLMs.
> > |Non-LLM|Aesthetic|WizardLM-13B|Aesthetic|openchat-3.5-0106|Aesthetic|
> > |-|-|-|-|-|-|
> > |MP|5.336 ± 0.010|ICL|5.515 ± 0.017|MP|5.479 ± 0.003|
> > |SFT|5.621 ± 0.025|ZO-OGD|5.635 ± 0.053|ZO-OGD|5.710 ± 0.081|
> > |Promptist|5.579 ± 0.006|AOZPT|**5.734 ± 0.028**|AOZPT|**5.828 ± 0.064**|
> >
> > 3. **Computational Efficiency**: Once the discrete prompt is generated, it is directly concatenated with task inputs and used for subsequent inference without further optimization or additional querying. Consequently, the mapping module **introduces no measurable inference latency**, and its cost is incurred only during the prompt-update steps (Table 13 in revised paper).
> >
> > >**Q1**:The paper uses frozen open-source LLMs to generate discrete prompts, but what if such LLMs are unavailable?
> >
> > If a frozen LLM is unavailable, AOZPT remains fully operational through two alternative mapping strategies:
> > 1. **Online-trained lightweight phrase–grammar optimizer**, which adapts to streaming data and learns to decode soft prompts into discrete text in real time.
> > 2. **Compact offline-trained adapter model**, which approximates the mapping function using a small set of prompt–response pairs.
> >
> > Both strategies maintain AOZPT’s online optimization loop and adaptive update mechanism. However, they **involve additional training**—either continuous online adaptation or offline supervised learning—which may introduce moderate computational or data-collection overhead.
> > Importantly, the frozen LLM is not a required dependency of AOZPT but rather a convenient, training-free option for implementing the soft-to-hard mapping. When available, a **frozen LLM provides a highly effective semantic decoder**: by leveraging its strong contextual reasoning and semantic generalization abilities, it produces discrete prompts that are coherent, robust, and well aligned with task objectives.

---

> > > ### Author Response · Authors · 2025-11-24
> > > **Responses to Reviewer DxWc-Part 3**
> > >
> > > > **Q2**:Can authors explain why there is no evaluation of the proposed method under adversarial noise in generative AI outputs ?
> > >
> > > We would like to clarify that **the noise considered in our paper corresponds to natural stochastic noise in generative model outputs, which differs fundamentally from adversarial noise**. As discussed in the manuscript (lines 34–38 in the revised manuscript, highlighted in blue), generative AI systems inherently exhibit randomness arising from sampling procedures, non-deterministic training elements, and variations in random seed initialization. These factors represent the realistic noise conditions under which online prompt optimization methods are typically applied.
> > >
> > > To assess AOZPT under such natural stochasticity, we conduct all experiments using three random seeds and report mean ± standard deviation across runs. The results (Table 2 in revised paper) show that AOZPT **not only achieves the the highest evaluation score across methods, but also maintains consistently small standard deviations**, indicating robustness to the inherent randomness of generative model outputs. For instance, AOZPT obtains 6.282±0.021 and 5.930±0.015 on the Anime dataset, and 6.656±0.015 and 6.313±0.009 on the Painting dataset. These results demonstrate that AOZPT remains stable and reliable under natural generative noise.
> > >
> > > Adversarial noise, however, represents a distinct setting involving intentionally designed perturbations aimed at misleading the model. Such scenarios are beyond the scope of the present study, which focuses on standard black-box online optimization. We agree that evaluating AOZPT under adversarially perturbed outputs is valuable and plan to explore this direction in future work.

---

> > > > ### Author Response · Authors · 2025-11-28
> > > > **Looking Forward to Further Discussion - Responses to Reviewer DxWc**
> > > >
> > > > We deeply appreciate your thoughtful review. We look forward to the opportunity to continue our discussion and further refine our work with your guidance.

---

### Official Review · Reviewer_rb5E · 2025-11-01

**Soundness:** 3
**Presentation:** 4
**Contribution:** 3
**Rating:** 6
**Confidence:** 2

**Summary:**

This paper introduces AOZPT, an online black-box prompt optimization method thta uses a two-point zeroth-order gradient estimator with an adaptive uncertainty scaling to counter both LLM output noise and ZO variance. It proves a sublinear local regret bound under noise and shows consistent empirical improvements across text2text and text2image tasks.

**Strengths:**

1) It is an interesting idea to move black-box prompt optimization from offline to online learning, this might be useful in training agentic systems that interacts with real environment.
2) The proposed method is clearly presented and very practical.
3) A mostly sound theory (maybe with some assumptions being a little bit too ideal) is built to support the effectiveness of the method.

**Weaknesses:**

1) Some assumptions, for example the Lipschitzness of \nabla f_t in z, can be too optimistic. Is it possible if the authors can further verify how much we can expect these assumptions to hold in practice, and when these assumptions fail to hold, how much impact does this have on the effectiveness of the method in practices?

2) I would recommend the authors to consider more baselines besides ZO-OGD. This can help isolating gains from the adaptive scaling agaisnt the two-point estimator.

**Questions:**

Please refer to Weakness).

---

> ### Author Response · Authors · 2025-11-24
> **Responses to Reviewer rb5E**
>
> We sincerely thank the reviewer for the positive evaluation of our work, including the recognition of the novelty of moving black-box prompt optimization to an online learning setting, the clarity and practicality of our method presentation, and the overall soundness of our theoretical analysis. Below, we provide detailed responses to the reviewer’s concerns, particularly the question about the practicality of assumptions.
> > **W1**:Some assumptions, for example the Lipschitzness of \nabla f_t in z, can be too optimistic. Is it possible if the authors can further verify how much we can expect these assumptions to hold in practice, and when these assumptions fail to hold, how much impact does this have on the effectiveness of the method in practices?
>
> Thank you for the reviewer’s thoughtful comment. We clarify our assumptions from three perspectives below:
> **1. Standard Theoretical Assumptions:** Assumptions 3.3–3.6 (Lipschitz gradient, bounded loss, bounded noise, bounded gradients) are standard and widely used in LLM optimization, online learning, and smoothness-based theoretical analysis[1,2,3,4,5,6,7]. These assumptions align with prior literature and are necessary for deriving meaningful guarantees.
>
> **3. Impact When Assumptions Fail:** If these assumptions are severely violated, the underlying LLM optimization typically becomes unstable: gradients may explode, the loss oscillates heavily, and training can diverge. **In such regimes, even standard training pipelines become unreliable, so a larger theoretical error bound is natural and reflects the inherent difficulty of learning rather than a limitation specific to our method**. Importantly, our experiments also include challenging scenarios with data drift (Table 5), where the loss landscape becomes highly non-smooth and the smoothness-related assumptions are more likely to fail. Even in these settings, our method consistently achieves the best empirical performance among all baselines, indicating that it is reasonably adaptive to deviations from the ideal assumptions. The use of multiple random seeds, and the observation that our method maintains its advantage under the additional noise (as shown by the reported mean ± standard deviation), further demonstrates its robustness when the assumptions are only approximately satisfied in practice.
>
> > **W2**: I would recommend the authors to consider more baselines besides ZO-OGD. This can help isolating gains from the adaptive scaling agaisnt the two-point estimator.
>
> We will include additional baselines beyond ZO-OGD to isolate the contributions from the adaptive uncertainty scaling. Specifically, we evaluate: (i) ZO-OGD with one-point estimator (only forward); (ii) AOZPT with one-point estimator (only forward) (Table 8 in revised paper). The results indicate that removing adaptive scaling leads to a substantial drop in performance, demonstrating that a large portion of the observed gains stems from the proposed scaling mechanism rather than solely from the two-point estimator itself.
>
> Table 8. Experimental results of the single-point method on the Anime and Painting datasets using the Stable Diffusion v1.5 model. $\Delta_1$ denotes the Adaptive Uncertainty Scale Adjustment mechanism and $\Delta_2$ denotes online zero-order prompt tuning.
>
> |Datasets|Methods|Aesthetic|
> |-|-|-|
> |Anime|AOZPT (single-point method) w/o $\Delta_1$ \& $\Delta_2$|5.710±0.021|
> ||AOZPT (single-point method) w/o $\Delta_1$ |5.815 ± 0.026|
> ||AOZPT (single-point method) |**5.872 ± 0.021**|
> |Painting|AOZPT (single-point method) w/o $\Delta_1$ \& $\Delta_2$|6.074±0.015|
> ||AOZPT (single-point method) w/o $\Delta_1$|6.184 ± 0.028|
> ||AOZPT (single-point method)|**6.219 ± 0.016**|
>
> [1] Zhan, Heshen, et al. "Unlocking black-box prompt tuning efficiency via zeroth-order optimization." Findings of the Association for Computational Linguistics: EMNLP 2024. 2024.
>
> [2] Malladi, Sadhika, et al. "Fine-tuning language models with just forward passes." Advances in Neural Information Processing Systems 36 (2023): 53038-53075.
>
> [3] Melo, Luckeciano C., Alessandro Abate, and Yarin Gal. "Stabilizing Policy Gradients for Sample-Efficient Reinforcement Learning in LLM Reasoning." arXiv preprint arXiv:2510.00819 (2025).
>
> [4] Seung, Hyunseok, Jaewoo Lee, and Hyunsuk Ko. "Low-Rank Curvature for Zeroth-Order Optimization in LLM Fine-Tuning." arXiv preprint arXiv:2511.07971 (2025).
>
> [5] Tan, Qitao, et al. "Harmony in divergence: Towards fast, accurate, and memory-efficient zeroth-order llm fine-tuning." arXiv preprint arXiv:2502.03304 (2025).
>
> [6] Shirkavand, Reza, et al. "Bilevel zofo: Bridging parameter-efficient and zeroth-order techniques for efficient llm fine-tuning and meta-training." arXiv preprint arXiv:2502.03604 (2025).
>
> [7] Liu, Yong, et al. "Sparse mezo: Less parameters for better performance in zeroth-order llm fine-tuning." arXiv preprint arXiv:2402.15751 (2024).

---

> > ### Comment · Reviewer_rb5E · 2025-11-28
> >
> > Thanks for the response. I tend to keep my original rating with increased confidence.

---

> > > ### Author Response · Authors · 2025-11-28
> > > **Thank You for Recognizing Our Work – Response to Reviewer rb5E**
> > >
> > > Thank you for your valuable feedback and for maintaining the original rating with increased confidence. We appreciate your insights, which have helped strengthen the clarity and confidence in our work. Your thoughtful review has been instrumental in improving the paper.

---

### Author Response · Authors · 2025-11-30
**Response to Area Chair:  Summary of Review Progress for Our Manuscript**

Dear Reviewers, Area Chairs, Senior Area Chairs, and Program Chairs:

We sincerely sorry for any inconvenience caused by the unforeseen events this year. We fully recognize the additional workload resulting from the recent OpenReview vulnerability and greatly appreciate the continued efforts of the project committee to uphold the integrity of the ICLR review process.

## Summary of the Current Review Progress
  * Reviewer rb5E, who initially gave a positive score of 6, has responded during the rebuttal period, providing positive and encouraging feedback and stating that they would “keep the original rating with increased confidence”.
  * Reviewers DxWc, nVJH, and wCmh have not yet engaged in further communication with us during the rebuttal period.

## Summary of Key Questions and Responses

We have summarized the key issues raised in the reviews and outlined the solutions we have implemented to address them:

1. **Validity and Practicality of the Assumptions** (Reviewers rb5E-W1, wCmh-W1)
   * We explained that while our assumptions are standard in LLM optimization, our method remains robust and effective even when these assumptions are partially violated, outperforming baselines in challenging scenarios (**Table 5**).
   * We replaced the uniformly bounded noise assumption with a more realistic sub-Gaussian noise model, which ensures similar theoretical results and demonstrates the extensibility of the assumption (**Appendix E.3**).
2. **State-of-the-art Baseline and More Task** (Reviewers DxWc, nVJH-W2, wCmh-W5)
   * We have added the SOTA baseline for reinforcement learning in online settings (**Table 17**).
   * We have also included an additional WMT/WMT14 De–En translation task (**Table 16**).
3. **Additional Ablations** (Reviewer rb5E-W2, Reviewer wCmh-W5)
   * We have conducted ablation studies with a one-point estimator beyond ZO-OGD to isolate the contributions of the adaptive scaling mechanism (**Table 8**).
   * We have conducted additional ablation studies on text-to-text tasks to further illustrate the contribution of each component of AOZPT (**Table 7**).
4. **Computational Overhead and Latency** (Reviewers DxWc-W1, nVJH-W3, wCmh-W4)
   * We clarified that AOZPT does not introduce additional inference latency in online environments, as it only updates the discrete prompt when new data arrives. The method requires similar or slightly more memory and latency compared to other methods like ICL, but does not incur unacceptable computational overhead (**Table 13**).
5. **Reliance on Open-Source LLMs** (Reviewers DxWc-W3, wCmh-W5)
   * We acknowledge that AOZPT relies on frozen open-source LLMs for prompt conversion, but this dependency is replaceable with smaller open-weight models or adapters. Additionally, the method introduces no measurable inference latency, as the discrete prompt is directly concatenated with task inputs and used for subsequent inference without additional computation.
   * We have included additional experiments using multiple open-source LLMs (**Table 14**).
6. **Noise Evaluation Under Natural Conditions** (Reviewers DxWc-Q2, wCmh-Q1)
   * We clarify that **the noise discussed in our paper corresponds to natural stochastic noise inherent in generative AI systems**, such as randomness from sampling procedures and variations in random seed initialization, not adversarial noise.
   * To assess the noise level empirically, we conduct experiments using three random seeds and report the mean ± standard deviation, demonstrating that AOZPT performs robustly under natural generative noise.
7. **Instruct and Larger Models** (Reviewer wCmh-Q2&Q3)
   * We explain that the **performance differences arise from differences in model capacity rather than limitations of AOZPT**. Additional experiments on GPT-4–class models (**Table 15**) show that AOZPT consistently provides complementary, model-agnostic improvements even when deployed on more powerful LLMs.
   * We clarify that all models in our experiments are already Instruct variants, and that discrepancies with official reported numbers stem from differences in evaluation protocols, including streaming-based online evaluation and prompting settings.

## Main Contribution of Our Paper
To address the **challenges of noise and high variance in online black-box prompt optimization**, our paper presents AOZPT method, an innovative approach that **integrates an uncertainty-scale adjustment mechanism**, enhancing both stability and performance  for LLMs in online learning settings.

We have addressed all the reviewers’ comments in our response and prepared a revised manuscript. Within the limited time available, some reviewers have not yet responded (Reviewers DxWc, nVJH, and wCmh), we respectfully request that the AC, SAC, and PC consider any score improvements warranted by our responses.

We express our sincere respect for your extra contributions!

Best Regards,

The Authors

---

### Meta-Review · Area_Chair_oyuH · 2026-01-11

**Summary:**

This paper studies black-box prompt optimization for generative AI systems in an online learning setting with noisy and stochastic outputs. The authors propose Adaptive Online Zeroth-order Prompt Tuning (AOZPT), which combines zeroth-order optimization with online learning and introduces an uncertainty-aware scaling mechanism to control noise and variance in non-convex optimization. The paper claims theoretical regret guarantees showing sublinear convergence and supports the analysis with empirical results demonstrating improved stability and performance over existing black-box prompt tuning methods in online scenarios.

**Reviewer Concerns:**

- Concern 1: Restrictive noise model assumption
Authors addressed by replacing the bounded-noise assumption with a sub-Gaussian noise model (Assumption 3.5 in revised Appendix E.3), demonstrating that theoretical guarantees extend to more realistic stochastic noise. Proof shows only bounded conditional second moments are required, making the framework strictly more general.

- Concern 2: Simulated vs. true online setting
The authors acknowledged the lack of publicly available streaming datasets for generative AI tasks. Implemented randomized streaming-from-URL protocol to simulate incremental data arrival. Partially addressed - remains simulated rather than true sequential user feedback, though authors provide reasonable justification given dataset limitations.

- Concern 3: Computational overhead not evaluated
Authors provided detailed runtime and memory analysis (Table 13). AOZPT shows ~3× training overhead vs. ICL but zero additional inference latency. Memory footprint is comparable across methods.

- Concern 4: Limited empirical validation
Authors substantially expanded experiments: (1) added multiple prompt-generating models (WizardLM-13B, OpenChat-3.5-0106), (2) reported full decoding configurations (temperature, sampling strategies), (3) added a machine translation task (WMT14), (4) clarified MP baseline = original model performance, (5) added text-to-text ablations (Table 7).

Concern 5: Performance comparison with larger models
Authors clarified Qwen2.5-14B (MP) > Llama-3.1-8B (AOZPT) reflects model capacity differences, not method limitations. Added GPT-4o-mini experiments (Table 15) showing AOZPT provides consistent improvements even on larger models, demonstrating method is complementary to model scaling.

Concern 6: Noise level quantification in practice
Authors quantified empirical noise through mean ± standard deviation across three random seeds (e.g., 6.282 ± 0.021 on Anime dataset), providing practical estimate of σ under autoregressive LLM variability. AOZPT maintains small standard deviations despite potential variance amplification.

**Reviewer Scores:**

Given the concerns of nVJH (Chengtao Jian), wCmh (Yunlong Hou) were adequately addressed; I believe they would be willing to increase the score.

---

### Decision · Program_Chairs · 2026-01-26

Accept (Poster)